# Oscillatory hyperactivity and hyperconnectivity in young *APOE*-ε4 carriers and hypoconnectivity in Alzheimer's disease

**Loes Koelewijn[1], Thomas M Lancaster[1,2,3], David Linden[1,2,3], Diana C Dima[1], Bethany C Routley[1], Lorenzo Magazzini[1], Kali Barawi[3], Lisa Brindley[1], Rachael Adams[1], Katherine E Tansey[4], Aline Bompas[1], Andrea Tales[5], Antony Bayer[6], Krish Singh[1,2]***

[1]Cardiff University Brain Research Imaging Centre, School of Psychology, Cardiff University, Cardiff, United Kingdom; [2]Neuroscience and Mental Health Research Institute, Cardiff University, Cardiff, United Kingdom; [3]MRC Centre for Neuropsychiatric Genetics and Genomics, Cardiff University, Cardiff, United Kingdom; [4]Core Bioinformatics and Statistics Team, College of Biomedical and Life Sciences, Cardiff University, Cardiff, United Kingdom; [5]Department of Psychology, College of Human and Health Sciences, Swansea University, Swansea, United Kingdom; [6]School of Medicine, Cardiff University, Cardiff, United Kingdom

**Abstract** We studied resting-state oscillatory connectivity using magnetoencephalography in healthy young humans (N = 183) genotyped for APOE-ε4, the greatest genetic risk for Alzheimer's disease (AD). Connectivity across frequencies, but most prevalent in alpha/beta, was increased in APOE-ε4 in a set of mostly right-hemisphere connections, including lateral parietal and precuneus regions of the Default Mode Network. Similar regions also demonstrated hyperactivity, but only in gamma (40–160 Hz). In a separate study of AD patients, hypoconnectivity was seen in an extended bilateral network that partially overlapped with the hyperconnected regions seen in young APOE-ε4 carriers. Using machine-learning, AD patients could be distinguished from elderly controls with reasonable sensitivity and specificity, while young APOE-e4 carriers could also be distinguished from their controls with above chance performance. These results support theories of initial hyperconnectivity driving eventual profound disconnection in AD and suggest that this is present decades before the onset of AD symptomology.
DOI: https://doi.org/10.7554/eLife.36011.001

*For correspondence:
singhkd@cardiff.ac.uk

**Competing interests:** The authors declare that no competing interests exist.

## Introduction

Once established, the cognitive impairments characteristic of Alzheimer's disease (AD) are irreversible (*Herrup, 2015*). To effectively identify measures to predict and prevent AD, it is vital to identify biomarkers at an early age, before the presence of any symptoms, and one increasingly popular route for this is neuroimaging of genetic risk groups. The greatest genetic risk factor for developing late-onset AD is the apolipoprotein E4 (*APOE*-ε4) allele (*Corder et al., 1993*; *van der Flier et al., 2011*).

*APOE* consists of 299 amino acids divided into receptor-binding and lipid-binding regions and has three isoforms (ε2, ε3 and ε4) that only differ at two of the amino acids (112 and 158), both of which are on the receptor-binding part of the protein. APOE has a crucial role in transporting cholesterol and other lipids, both in plasma and in the brain, but it is its role in regulating amyloid-β that

has received much attention, due to the latter's apparently crucial role in AD (*Reinvang et al., 2013*; *Zhao et al., 2018*). The pathological accumulation of amyloid-β could occur via multiple mechanisms of over-production and/or impaired clearance (*Zhao et al., 2018*) and it appears that the presence of the APOE-ε4 isoform may differentially lead to one or more of these occurring. For example, compared to APOE-ε3, APOE-ε4 stimulates increased recycling of the amyloid precursor protein (APP), hence leading to an increase in amyloid-β production (*Ye et al., 2005*). This mostly appears to be via modified interactions between APOE-ε4 and the LRP1 receptor. Similarly, it has been shown that APOE can bind to amyloid-β to form a complex that then promotes clearance via cell internalization of this complex (*Zhao et al., 2018*). As the APOE-ε4-amyloid-β complex is less stable than the other isoforms, this may make it less effective in amyloid clearance. Finally, APOE-ε4 may interfere with protease-mediated clearance of amyloid-β via downregulation of, for example, insulin-degrading-enzyme (IDE) (*Du et al., 2009*).

Whatever the precise mechanism, APOE-ε4 carriers have a much increased risk of developing sporadic AD (*Corder et al., 1993*), of progression to AD if diagnosed with mild cognitive impairment (MCI) (*Petersen, 1995*), and to develop AD at an earlier age (*Corder et al., 1993*; *van der Flier et al., 2011*) compared to non-carriers. Studying early effects of the *APOE-ε4* allele on the brain may therefore reveal insights into the neural pathways leading to AD.

Within groups of AD patients and elderly controls, *APOE*-ε4 carriers show alterations in brain structure and function that follow similar patterns to those that characterise AD, but are stronger than in non-carriers. Imaging studies show that *APOE*-ε4 carriers present with more severely atrophied temporal areas (*Filippini et al., 2009a*; *Filippini et al., 2009b*; *Juottonen et al., 1998*), and have elevated levels in temporo-parietal areas of the two main harmful brain deposits seen in AD: the tau protein (*Ossenkoppele et al., 2016*), causing neurofibrillary tangles, and amyloid-β (*Drzezga et al., 2009*), the peptide accumulating in the pathognomonic intercellular plaques.

Functional neuroimaging during a 'resting state' can show intrinsic fluctuations in levels of background brain activity. Fast fluctuations of rhythmic neural activity occur spontaneously and are thought to underlie communication between brain areas (*Schnitzler and Gross, 2005*). These neural oscillations, particularly in frequency bands such as the alpha band, have been linked to patterns of white matter connectivity (*Teipel et al., 2009*). *APOE* plays a role in lipid transport, important for myelination of white matter (*Herrup, 2015*). Studying neural oscillations in conjunction with variation in this gene may therefore reveal important changes in neural connectivity. Neural oscillations can be non-invasively measured in vivo in humans using electro- or magnetoencephalography (E/MEG), which measure fluctuations in electric potentials or magnetic fields and hence directly reflect neural activity at the scalp with high temporal resolution (*Baillet, 2017*).

To date, M/EEG studies in elderly *APOE*-ε4 carriers have shown changes in the oscillatory alpha band (*Babiloni et al., 2006*; *Canuet et al., 2012*; *de Waal et al., 2013*; *Jelic et al., 1997*; *Kramer et al., 2008*; *Ponomareva et al., 2008*) and in the much lower-frequency delta band (*Cuesta et al., 2015*; *de Waal et al., 2013*) in regions similar to those reduced in AD and MCI. These findings suggest that the presence of an *APOE*-ε4 allele is associated with differences in oscillatory brain function at a stage preceding AD symptomology, but it is not clear how early in the life span this is established. One very small-sample study has measured MEG oscillations in healthy young genotyped individuals, and found that low-frequency (theta) oscillations were increased in the medial frontal cortex of *APOE*-ε4 carriers, but did not investigate the alpha band, nor connectivity between brain regions (*Filbey et al., 2006*).

Here, we used MEG to study resting-state neural activity non-invasively in a large sample of healthy young individuals who were genotyped for *APOE* alleles (51 ε4 carriers and 108 non-carriers). We assessed whether and how background oscillatory neural activity and connectivity across the whole brain in a range of frequencies is affected in young *APOE*-ε4 carriers, decades before the effects of ageing or potential onset of AD symptomology. To relate the significance of these findings to brain dysfunction in AD, we further compared the young *APOE*-ε4-related findings with resting-state MEG data from patients with established AD and healthy age-matched controls. Finally, using machine learning approaches, we assessed the diagnostic information content of these MEG-derived static connectivity matrices by quantifying their accuracy in solving three problems: 1) Whether young *APOE-ε4* carriers could be distinguished from young non-carriers, 2) Whether AD patients could be distinguished from elderly controls and 3) Whether a classifier trained on identifying young E4 carriers could identify AD patients, purely from their MEG connectivity matrices.

# Results

## Participants

*APOE*-ε4 carriers did not significantly differ from non-carriers in age, gender, years of education, hippocampal volume, head motion during the recording or number of good-quality resting-state data epochs included in the analysis (*Table 1*). Performance differed on only 1 out of 8 cognitive tasks, the NAB Mazes task (reasoning and problem solving), where ε4 carriers performed better than non-carriers, but this result (p=0.03) did not survive a false-discovery-rate correction for multiple comparisons (*Benjamini and Yekutieli, 2001*). Similarly, within the 68 regions of the Desikan-Killiany gyral-based atlas, there was no significant difference found in cortical thickness, volume or area, in any region, when corrected for multiple comparisons.

## *APOE*-ε4 carrier versus non-carrier group differences in oscillatory connectivity

*Figure 1* shows differences in static oscillatory network connectivity for *APOE*-ε4 carriers, compared to controls. In the top row, it is clearly evident that, for the individual frequency bands, the beta band has the greatest number of valid connections (224) and the highest number of edges (n = 13) that exceed our initial thresholding criterion of p<0.05 (uncorrected). All of these edges show increased connectivity in the beta band.

In the *Combined* maps, there are not only more edges than any of the individual bands (n = 282) but this synthesised vector combination also shows enhanced sensitivity to cohort effects, with more connections found to be significantly different at the edge-level (n = 31), the cluster-size level

**Table 1.** Participant demographics and statistical assessment of APOE-ε4 group differences (two-sample t-tests).

| | All MEG datasets | *APOE*-ε4 carriers | *APOE*-ε4 non-carriers | P value |
|---|---|---|---|---|
| Number | 183 | 51 | 108 | |
| Male/female | 60/123 | 18/33 | 36/72 | 0.81 |
| Allele composition | | 41 ε3ε4<br>6 ε4ε4<br>4 ε2ε4 | 87 ε3ε3<br>17 ε2ε3<br>4 ε2ε2 | |
| Age[*] | 24.5 ± 5.4 years | 25.2 ± 6.8 years | 24.2 ± 4.6 years | 0.23 |
| Years of education | | 16.4 ± 0.9 (N = 15) | 17.0 ± 1.3 (N = 34) | 0.13 |
| Cognitive tasks:[†]<br>Trail making test<br>BACS symbol coding<br>HVLT-R (verbal learning)<br>WMS-III SS (spatial span)<br>Letter-number span<br>NAB mazes (reasoning)<br>BVMT-R (visual learning)<br>Category fluency | | 27.8 ± 10.1 (N = 47)<br>65.0 ± 8.7 (N = 48)<br>27.7 ± 3.9 (N = 48)<br>16.9 ± 3.4 (N = 47)<br>15.0 ± 2.9 (N = 48)<br>22.9 ± 3.0 (N = 48)<br>28.3 ± 5.5 (N = 48)<br>29.2 ± 6.8 (N = 47) | 27.1 ± 8.8 (N = 98)<br>66.3 ± 9.1 (N = 100)<br>27.2 ± 4.3 (N = 100)<br>16.5 ± 3.5 (N = 99)<br>15.1 ± 2.8 (N = 100)<br>21.4 ± 4.3 (N = 100)<br>27.8 ± 5.1 (N = 100)<br>28.6 ± 6.9 (N = 100) | 0.65<br>0.42<br>0.51<br>0.51<br>0.86<br>0.03<br>0.54<br>0.60 |
| Hippocampal volume (L) absolute, relative[‡] | 4343 ± 408 mm$^3$<br>2.734 ± 0.26 | 4362 ± 422 mm$^3$<br>2.705 ± 0.24 | 4228 ± 394 mm$^3$<br>2.746 ± 0.27 | 0.74<br>0.35 |
| Hippocampal volume (R) absolute, relative[‡] | 4439 ± 368 mm$^3$<br>2.797 ± 0.25 | 4417 ± 363 mm$^3$<br>2.757 ± 0.23 | 4416 ± 345 mm$^3$<br>2.798 ± 0.26 | 0.67<br>0.35 |
| Total ICV | 1596 ± 161 cm$^3$ | 1618 ± 156 cm$^3$<br>(N = 50) | 1589 ± 166 cm$^3$<br>(N = 106) | 0.29 |
| MEG head motion | 0.28 ± 0.23 cm | 0.32 ± 0.26 cm | 0.27 ± 0.21 cm | 0.18 |
| MEG number of epochs | 144.75 ± 7.63 | 144.25 ± 7.02 | 145.06 ± 7.75 | 0.53 |

[*]Values are represented as group mean ±SD.

[†]The MCCB includes tests of processing speed (BACS symbol coding, category fluency: animal naming, trail making test), working memory (WMS-III: spatial span, letter-number span), verbal and visual learning, and reasoning and problem-solving skills (NAB mazes).

[‡]Relative left (L) and right (R) hippocampal volumes are calculated as absolute hippocampal volumes divided by total intracranial volume (ICV), multiplied by 1000.

DOI: https://doi.org/10.7554/eLife.36011.002

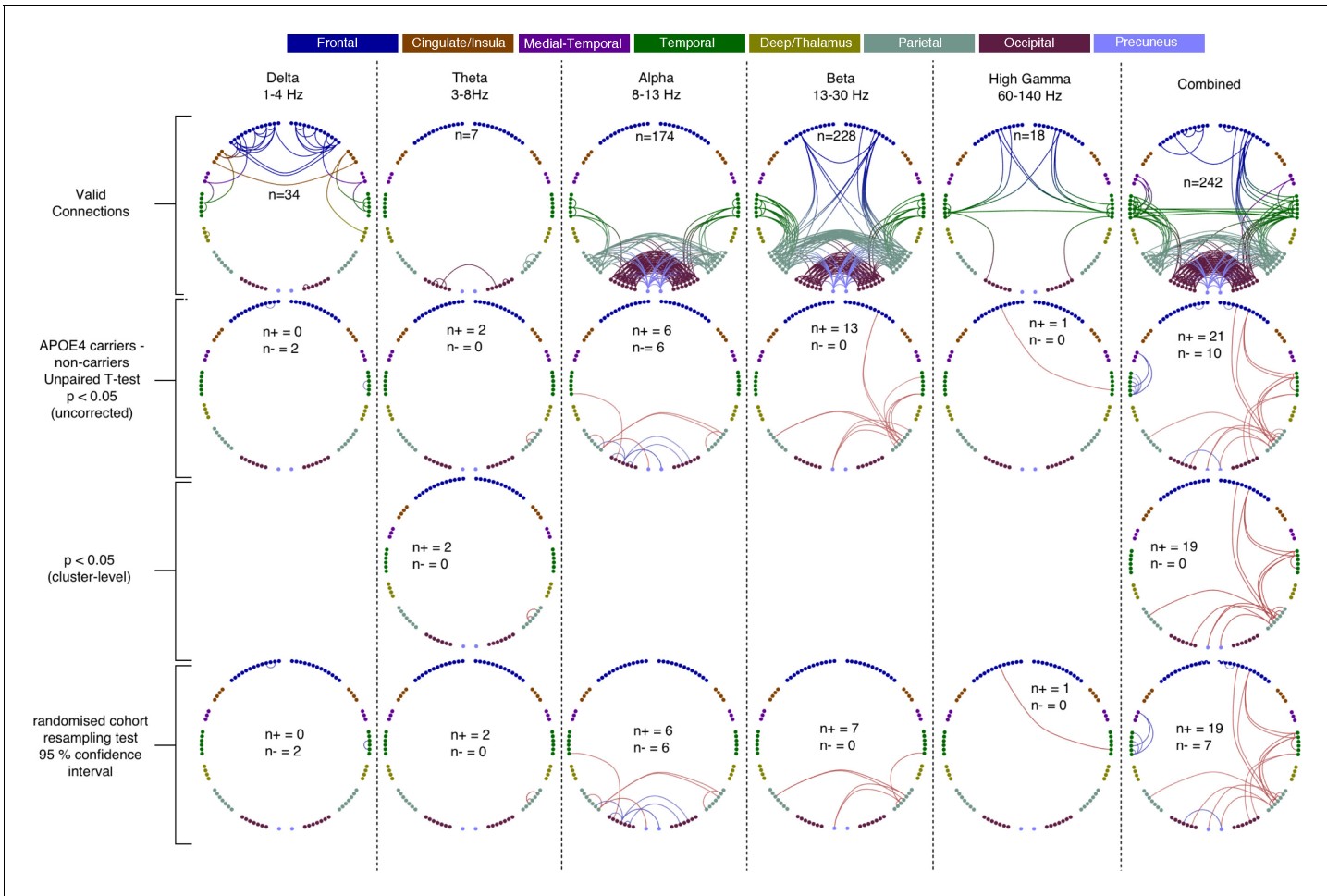

**Figure 1.** Each column shows oscillatory amplitude correlations comparing APOE-ε4 carriers and non-carriers for each of the five frequency ranges where valid edges were found, and combined across frequency. The top row shows where there are 'valid' edge connections in either of the two groups. Colours, with a key at the top of the figure, indicate regions of brain areas that the connections originate from. The numbers on the plots show the number of valid edges found. Note that Low Gamma is not shown as no 'valid' edges were found. The second row shows the unpaired t-statistic for APOE-ε4 carriers compared to non-carriers. Increases are displayed in red and decreases in blue and only those connections significant at p<0.05 (uncorrected) are shown. The numbers on each plot (n+/n-) show the number of significant increases and decreases found in each map. Note that, across all tests, only one edge connection, in the Theta band, was found to be significantly different when corrected for multiple comparisons (p<0.036, right parietal). Using the cluster-level network-based size statistic, shown in the third row, significantly large clusters were found for the theta and Combined maps. The bottom, fourth row, shows the results of the cohort random sampling analysis. Edges are only shown if they are within the 95% confidence limit for APOE-ε4 -related increases (red) or decreases (blue). Note that, in general, the predominant effect is for edges to show increases in connectivity in APOE-ε4 carriers, compared to non-carriers and that this is most clearly shown in the Combined map (right-hand column), in which the most valid connections are identified and the most significant, and consistent, differences are seen. In all connection plots, line opacity indicates relative connection strength within the plot (most opaque = strongest). Data and scripts needed to generate this figure can be found at http://dx.doi.org/10.17605/OSF.IO/E4CJX.

DOI: https://doi.org/10.7554/eLife.36011.003

(n = 19) and after cohort resampling tests (n = 26). Again, the balance of the sign of the effect is skewed towards hyperconnectivity in the majority of comparisons, although, interestingly, there is some evidence of hypoconnectivity (blue lines) in left medial temporal regions, including the hippocampus.

What can also be observed is a bias towards effects in the right hemisphere, with many of the significantly different edges arising from the right parietal cortex, but also including medial precuneus regions. These differences therefore at least partially overlap with posterior regions of the human default-mode network (DMN). Finally, what is encouraging is that most of these significant differences are observable in the 95% confidence interval test using random sub-sampling of the cohort

(bottom row), suggesting that our results may well generalise to the wider population, rather than being driven by a small number of cases, or controls, in our sample.

## APOE-ε4 carrier versus non-carrier group differences in oscillatory activity

When we assessed activity across the brain using a voxelwise assessment of temporal variability in the amplitude envelope at each location in the brain, the only differences found using TFCE (p<0.05 corrected) were in the two gamma ranges, 40–60 Hz and 60–140 Hz and are shown in *Figure 2*. These were exclusively gamma hyperactivities in a right-lateralised profile, distributed over lateral parietal, temporal and frontal regions. When a more conservative voxel-level threshold (p<0.05, corrected) was applied, only one region survived in the 60–140 Hz range, localised over the right middle and inferior frontal gyri.

In summary, gamma hyperactivities were found in mostly right-lateralised regions, some of which also appear to show hyperconnectivity in the lower frequency ranges (principally beta and alpha). This overlap is described graphically in the lower part of *Figure 2*.

## Alzheimer's disease versus elderly controls

Here, we compared our finding of a spatially-localised increase in functional brain connectivity in young *APOE*-ε4 carriers to connectivity changes in a cohort of AD patients, in which we have recently reported strongly decreased alpha and beta-band oscillatory network activity in parieto-temporal areas with MEG (*Koelewijn et al., 2017*), for AD patients compared to age-matched controls. In our reanalysis of a subset of the same dataset, using the current pipeline (*Figure 3*) to compare AD patients to age-matched controls, we show strongly decreased oscillatory connectivity in predominantly parieto-temporal connections, with the most significant, and extensive, effects visible in the Alpha and Beta ranges. Again, similar to the young *APOE*-ε4 analysis, the most valid connections, and the most extensive pattern of significant differences, was observed in the synthesised vector combination of frequency-specific effects (right hand column). We also note that, in contrast to the *APOE*-ε4 analysis, most of the observed significant cohort differences are reduced connectivity, i.e. *hypoconnectivity,* in AD compared to elderly controls.

Finally, when using the temporal standard-deviation of the oscillatory envelope to quantify activation at each AAL node, this cohort of patients did not show any significant differences, in any frequency band, compared to their age-matched controls.

## Comparing the effects in young APOE-ε4 carriers to elderly patients with Alzheimer's disease

*Figure 4* shows a comparison of effects, taken from the most sensitive *Combined* connectivity maps, in order to directly visualise the overlap of statistical differences in the two experiments. In terms of the connections identified as valid (top row of *Figure 1*, *Figure 3* and *Figure 4a*), there are clear similarities but also differences, principally in terms of the balance between parieto-occipital connectivity and parieto-temporal-frontal connections. This may reflect differences due to the large discrepancy in age between the two experimental cohorts. In terms of cohort effects, it can clearly be seen that in the *APOE*-ε4 comparison, young carriers mostly demonstrate hyperconnectivity in a right dominant network, whilst in the AD comparison, more extensive bilateral decreases in connectivity are seen. In *Figure 4c* we show this in more detail by displaying the unthresholded t-statistics for each comparison and then comparing the magnitude of effect-sizes for the two groups in the scatter plot. There is a clear negative correlation of effect sizes between the two experiments (r = −0.18, p<3.6×10$^{-6}$), confirming the fact that, on average, connections that show hyperconnectivity in *APOE*-ε4 carriers demonstrate hypoconnectivity in Alzheimer's cases. However, there is clearly not a perfect correspondence between the pattern of effects.

We can also perform a more conservative comparison of the effects in the two experiments by restricting our analysis to those connections, and AAL regions, that show a significant effect in both experiments. This is shown in *Table 2*. In the Alpha band there is a single occipital connection that has reduced connectivity in both *APOE*-ε4 carriers and AD. In the Beta band there is a single connection from the medial precuneus region to right lateral parietal that shows hyperconnectivity in *APOE*-ε4 carriers and hypoconnectivity in people with AD, compared to their respective controls.

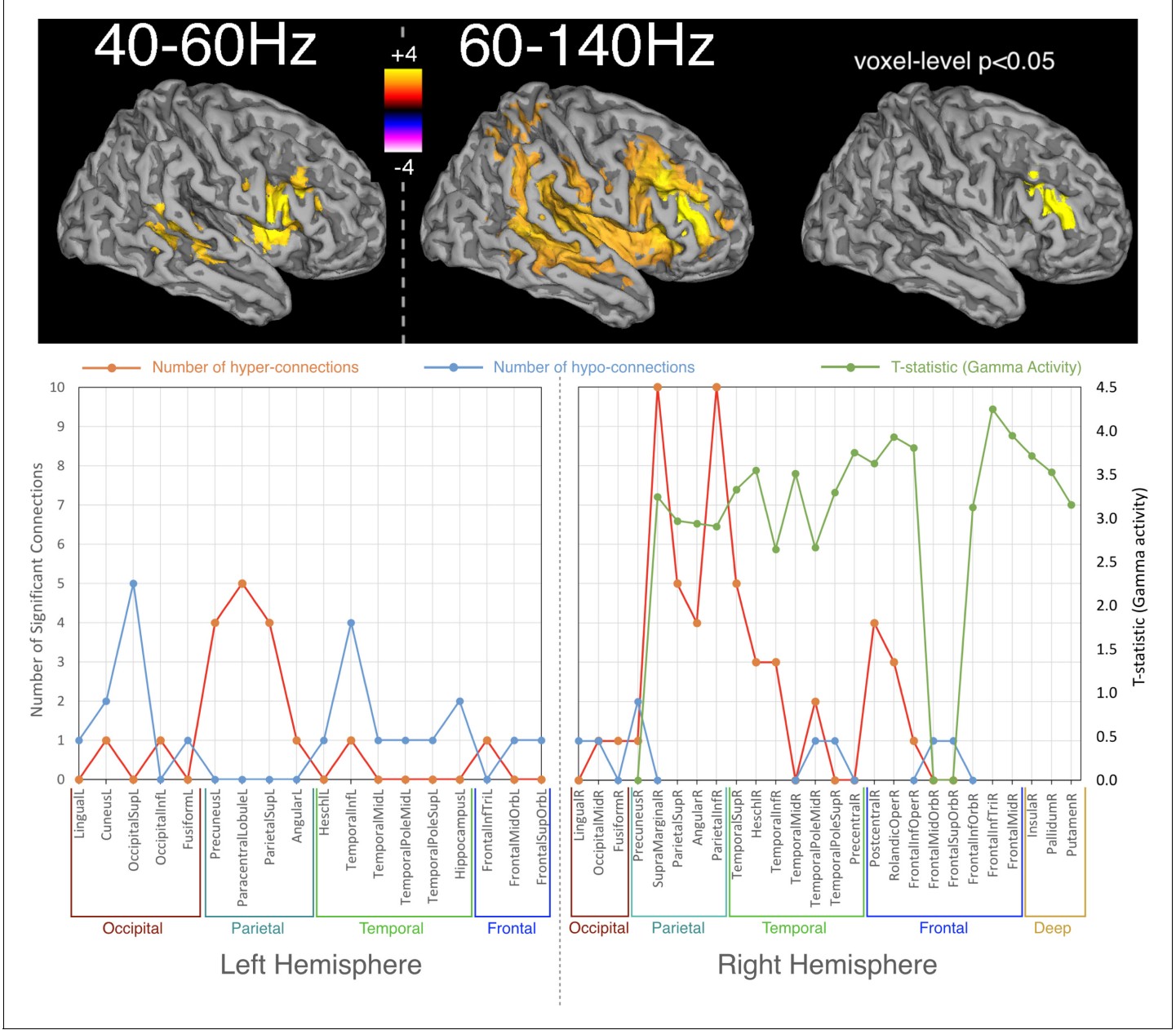

**Figure 2.** Spatial distribution of differences in oscillatory activity, between young APOE-ε4 carriers and their matched controls and the relationship with hyper- and hypo- connectivity. Top panel shows differences in activity as a coloured overlay depicting unpaired t-statistics (thresholded at p<0.05, corrected using TFCE), with yellow/orange representing greater activity in the APOE-ε4 carriers. Significant effects were only found in the right hemisphere and only for the two gamma ranges tested, 40–60 Hz and 60–140 Hz. The overlays are shown depicted on a template brain surface. The far-right column shows the results for the 60–140 Hz comparison using a more stringent voxel-wise correction for multiple-comparisons. The bottom panel shows the distribution of differences in both connectivity and activity, between APOE-ε4 carriers and non-carriers, within each AAL node that shows any significant effect. For clarity the left and right hemispheres are shown separately. Red lines show the number of significant connections to/from each node that are of higher strength in APOE-ε4 carriers, compared to non-carriers. Blue lines show the number of significant connections to/from each AAL node that are weaker in APOE-ε4 carriers. For the right hemisphere only, as effects were only found on the right, the green lines show the most significant t-statistic found in each AAL region for the analysis of gamma activity. Positive values indicate that gamma activity, in either band, was higher in APOE-ε4 carriers compared to non-carriers. Note that connection numbers are the sum across all frequency ranges tested, with most of these coming from the low frequencies – only one connection (FrontalInfTriL to TemporalPoleMidR) showed hyperconnectivity in the gamma range. In contrast, only gamma showed hyperactivity, so the green lines/marker exclusively reflect right-hemisphere gamma hyperactivity. The spreadsheet used to generate the graphs on this figure can be found at http://dx.doi.org/10.17605/OSF.IO/E4CJX.
DOI: https://doi.org/10.7554/eLife.36011.004

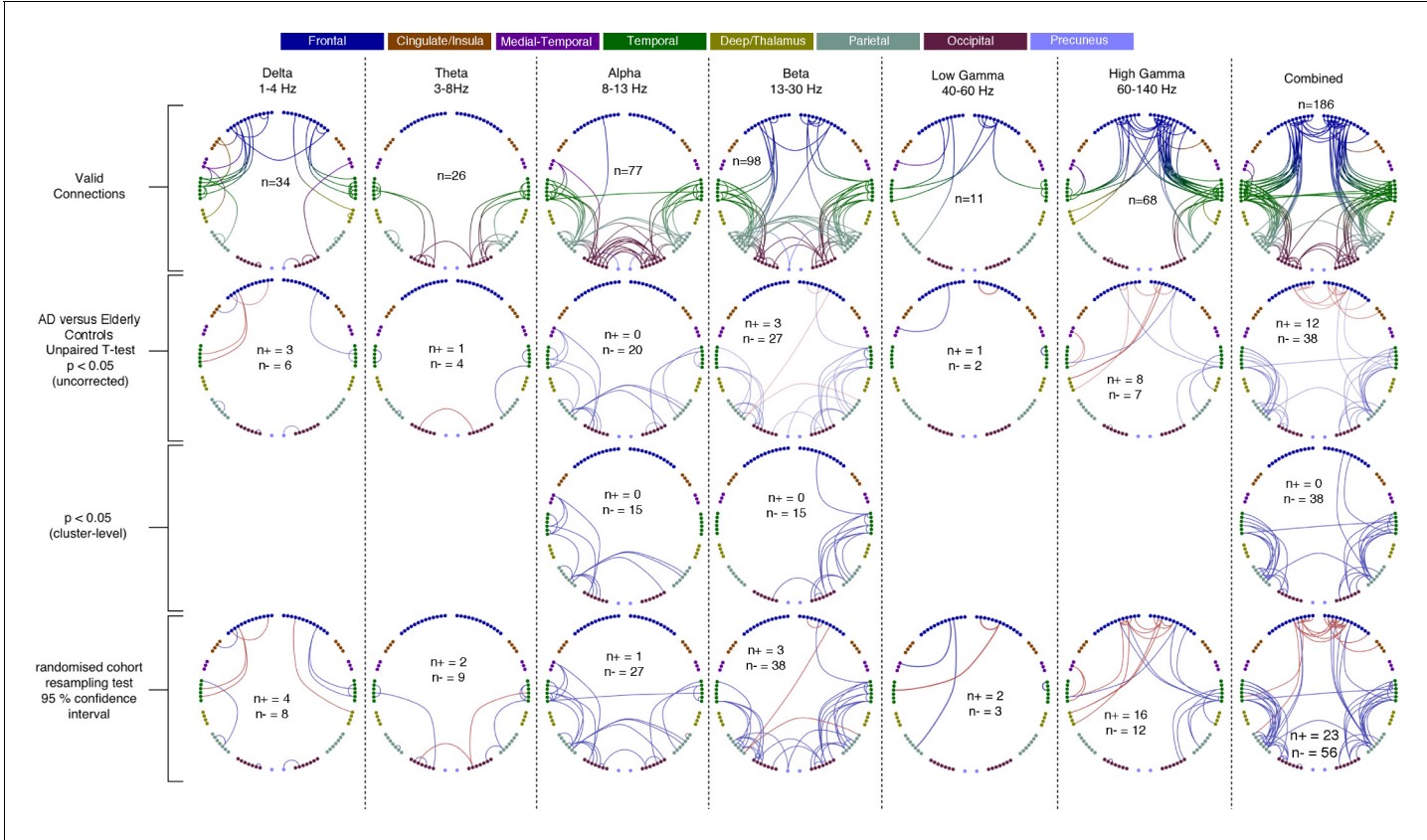

**Figure 3.** Each column shows oscillatory amplitude correlations comparing AD patients and matched elderly controls for each of the six frequency ranges where valid edges were found, and combined across frequency, and the Combined map. The top row shows where there are 'valid' edge connections in either of the two groups. Colours, with a key at the top of the figure, indicate regions of brain areas that the connections originate from. The numbers on the plots show the number of valid edges found. The second row shows the unpaired t-statistic for AD compared to controls - Increases are displayed in red and decreases in blue and only those connections significant at p<0.05 (uncorrected) are shown. The numbers on each plot (n+/n-) show the number of significant increases and decreases found in each map. In terms of corrected significance only one connection each reached the p<0.05 threshold in Beta, High-Gamma and Combined (not shown). Using the cluster-level network-based size statistic, shown in the third row, significantly large clusters were found for the Delta, Alpha, Beta and Combined maps. The bottom row shows the results of the cohort random sampling analysis. Edges are only shown if they are within the 95% confidence limit for AD-related increases (red) or decreases (blue). Note that, in general, the predominant effect is for edges to show decreases in connectivity in AD patients, compared to controls, and that this is most clearly shown in the Combined map (right-hand column), in which the most valid connections are identified and the most significant, and consistent, differences are seen. Data and scripts needed to generate this figure can be found at http://dx.doi.org/10.17605/OSF.IO/E4CJX.
DOI: https://doi.org/10.7554/eLife.36011.005

Similarly, in the *Combined* analysis, there are three connections in the right lateral-parietal/temporal region that show this same pattern of effects.

## Machine-learning results

*Table 3* shows the cross-validated results of the support-vector-machine (SVM) classification tests. Using pooled classification, based on valid edges in all frequency ranges, the classifier was, on average, able to distinguish AD patients from elderly controls with reasonable specificity and sensitivity and a significant AUC of 77.6% (p<0.02). Performance classifying *APOE-ε4* carriers from non-carriers was also significantly above the chance level of 50%, with an AUC of 63.5% (p<0.005). Interestingly, when we trained a classifier to distinguish *APOE-ε4* carriers, using all data from the young cohort, and then predict AD cases from controls, performance was significantly below chance, with an AUC of 31.8%. If we assume that the classifier is predicting reverse labelling in the disease group, this was equivalent to an AUC of 68.2% that is above chance, but only a trend to significance (p<0.14). However, this apparent reversed labelling reinforces the principal findings of this paper

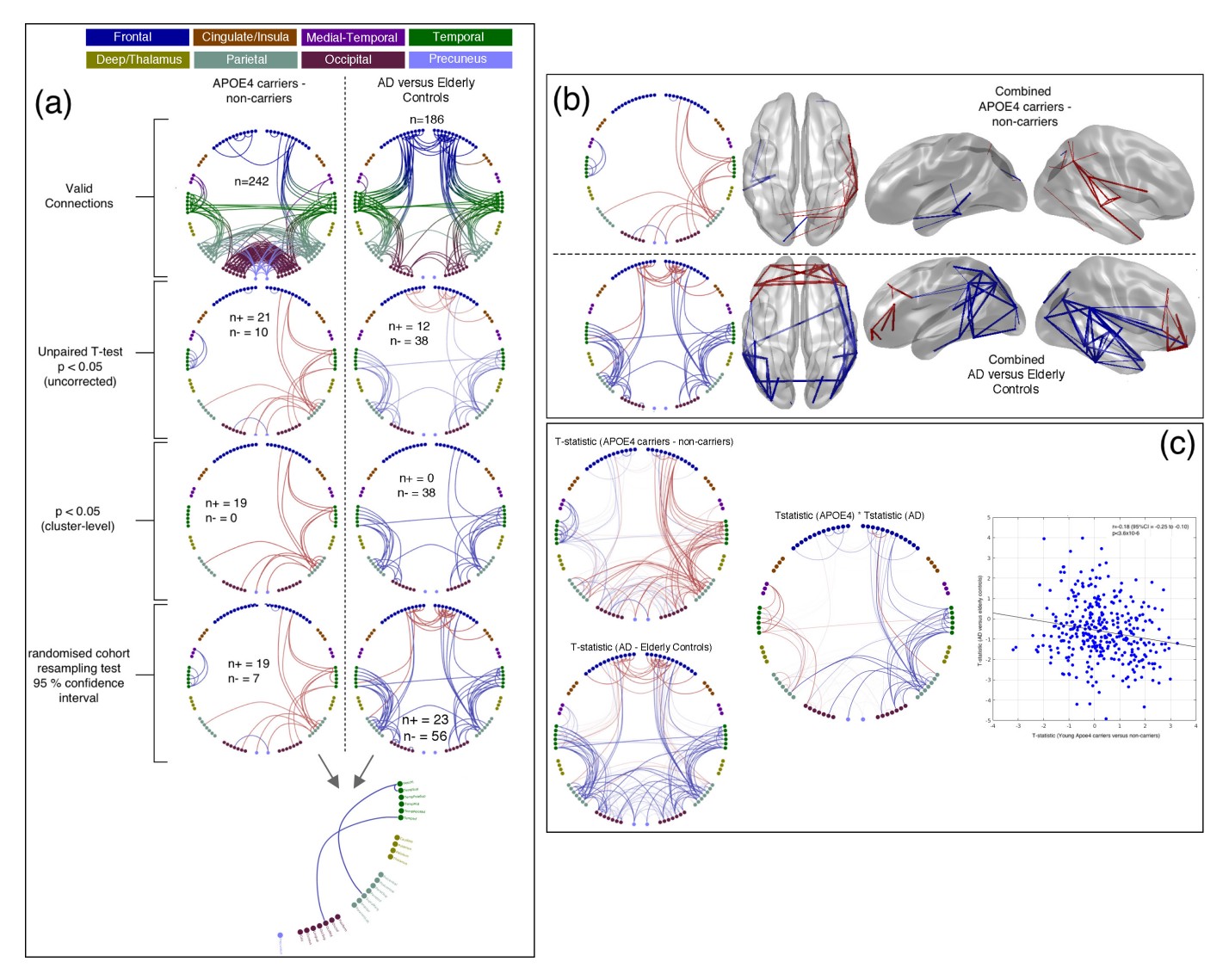

**Figure 4.** Comparison of connectivity differences in the two experiments, assessed using the Combined connectivity maps. (a) Replotting of the results for the Combined maps, extracted from *Figure 1* and *Figure 2* and displayed in the same format. At the bottom of this panel, a conjunction analysis is shown in which connections that meet the 95% confidence criterion for both experiments are displayed. Three connections survive this test and are shown in blue to reflect the fact that the effects have opposite sign (see *Table 2* for more details). (b) A comparison of connections surviving a 95% confidence interval threshold, estimated using cohort resampling, for the APOE-ε4 versus young controls experiment (top row) and the AD versus elderly controls experiment (bottom row). Here we display the thresholded connections on both the circle plots (left column) and three views of a template brain (columns 2–4). (c) A direct comparison of effect sizes. In the left column, unthresholded t-statistics are displayed for the two experiments, with the opacity of the lines reflecting the magnitude of the t-statistic. In the middle, the two sets of t-statistics are multiplied together for display. In this panel, strong lines represent connections with high t-statistics for both the APOE-ε4 experiment and the AD experiment, with blue colours representing connections which show opposite effect directions for example hyperconnectivity in APOE-ε4 and hypoconnectivity in AD. In the right panel, the scatter plot shows the effect-size (i.e. unpaired t-statistic, thresholded at p<0.05 uncorrected) for the AD versus elderly control experiment, plotted against the effect-size for the APOE-ε4 carriers versus non-carriers experiment. A clear negative relationship is observed. Data and scripts needed to generate this figure can be found at http://dx.doi.org/10.17605/OSF.IO/E4CJX.
DOI: https://doi.org/10.7554/eLife.36011.007

that *APOE-ε4* carrier status in young controls is principally revealed by *hyperconnectivity*, whilst Alzheimer's disease leads to *hypoconnectivity* in a larger, but overlapping, set of connections.

**Table 2.** Connections which show a statistically significant effect in both experiments.

| Frequency Range | AAL Region 1 | AAL Region 2 | E4 carriers versus non-carriers (T-statistic) | AD versus elderly controls (T-statistic) |
|---|---|---|---|---|
| Alpha | Cuneus left | Lingual left | −2.5 | −1.8 |
| Beta | Parietal superior right | Precuneus left | +2.3 | −2.3 |
| Combined | Occipital mid right | Temporal inferior right | +2.3 | −2.2 |
| Combined | Supramarginal right | Heschl right | +2.5 | −3.2 |
| Combined | Temporal Superior right | Heschl right | +3.0 | −3.0 |

DOI: https://doi.org/10.7554/eLife.36011.006

## Discussion

In a large sample of young healthy *APOE-ε4* carriers, and using a conservative approach in which we restrict our analysis to highly reproducible (top 20%) connections, we found consistent increases in resting-state oscillatory activity and connectivity in a right-dominant network of parieto-occipital, parieto-temporal and parieto-frontal connections. The largest and most consistent cluster of effects arose in the beta-band, representing rhythmic synchronized activity thought to reflect global input at the synapse (*Schnitzler and Gross, 2005*) and in our analysis of Combined connectivity maps. These findings occurred in absence of cognitive or structural brain differences. We also showed that those connections showing the largest average increases in connectivity tended to overlap with those connections that showed decreased parieto-temporal connections in AD patients, and was most profound in connections to/from right-hemisphere parietal areas. Our results therefore suggest that the right parietal cortex is a particularly important area in the risk for, and development of, AD. Although our study can only show an implied association between early hyperconnectivity and later hypoconnectivity, we could speculate that an initial hyperconnectivity/hyperactivity centred around (right) parietal cortex has a cascade of effects that may lead to eventual disconnection in AD. Although initially a right-dominant effect, this may spread over time to more widespread effects across the cortex and hemispheres. A larger study, using the methods we present here, over a wider age-range, may shed further light on this.

Interestingly, when analysing activity differences in oscillatory envelopes, rather than connectivity, the only significant differences found were *APOE-ε4* related hyperactivity in the gamma bands, both 40–60 Hz and 60–140 Hz. These gamma activity increases overlapped with those right-hemisphere brain regions showing hyperconnectivity in the lower bands (alpha/beta). This result is consistent with theories suggesting that gamma band oscillations mostly reflecting local, rather than global processing, whilst alpha/beta underpin longer-range interactions (*Donner and Siegel, 2011*). Our results are also consistent with a recent computational model in which it was shown that a realistically-coupled set of local gamma oscillators, at multiple simulated brain regions, led to long-range, slowly varying, coupled amplitude-envelopes, but in the alpha/beta range (*Cabral et al., 2014*). Finally, it has been suggested that higher-gamma band activity, such as our 60–140 Hz band, may be correlated with neuronal spiking rates (*Ray and Maunsell, 2011*). If true then the higher gamma-

**Table 3.** Leave-one-out Cross-validated performance of within experiment (top two rows) and cross-experiment (bottom row) SVM classification tests.

| Problem | Performance using all valid features |
|---|---|
| Classify AD versus elderly controls eyes open | Specificity/Sensitivity = 76.7 % / 65.3% AUC = 77.6%, p<0.019 |
| Classify E4 versus nonE4 eyes open | Specificity/Sensitivity = 56.8 % / 58.6% AUC = 63.5%, p<0.005 |
| Train on E4 versus nonE4 then classify AD versus elderly controls eyes open | Specificity/Sensitivity = 36.4 % / 28.6% AUC = 31.8% (68.2% using reversed labels) p<0.14 |

DOI: https://doi.org/10.7554/eLife.36011.008

band activity we see in APOE-ε4 carriers may be a reflection of the neuronal hyperactivity, associated with increased amyloid-β, seen in rodent AD models (*Stargardt et al., 2015*).

The highly specific spatial patterns of increased resting-state oscillatory connectivity in young APOE-ε4 carriers, identifying the right parietal cortex as the most affected area, including inter-hemispheric connections and connections to the precuneus, complement and extend current neurophysiology findings. MEG studies using Graph Theory network measures have also identified a loss of connectivity in AD, with the parietal cortex as the most affected hub (*de Haan et al., 2012b*; *Tijms et al., 2013*). Effects were found specifically in the beta band – thought to play an important role in long-range functional connectivity (*Schnitzler and Gross, 2005*). Furthermore, previous EEG studies on APOE-ε4 have identified differences in power or connectivity in the alpha band, but these have only been investigated in older participants (*Babiloni et al., 2006*; *Canuet et al., 2012*; *de Waal et al., 2013*; *Jelic et al., 1997*; *Kramer et al., 2008*; *Ponomareva et al., 2008*). Using the higher spatial resolution of MEG, Cuesta et al. (*Cuesta et al., 2015*) studied source-localised connectivity in elderly participants with and without MCI, and found that APOE-ε4 carriers showed differences only in a much lower frequency band than alpha (delta, 1–4 Hz), focused around a frontal brain hub. Cuesta et al. measured phase-based connectivity, using the phase-locking value, a very different measure than the amplitude correlations we used, and more prone to source leakage artefacts (*Colclough et al., 2016*). Interestingly, in our previous study we found clear changes in oscillatory amplitude, but not phase correlations, in the brains of AD patients (*Koelewijn et al., 2017*), though we did not investigate MCI patients. For these reasons, and to reduce multiple comparisons, we did not investigate phase-based connectivity here.

To our knowledge, only one small MEG study (N = 8) has previously investigated young APOE-ε4 carriers, and found increases in mainly medial-frontal theta-band (3.5–7 Hz) oscillatory power during a working memory task (*Filbey et al., 2006*). This study did not investigate connectivity, nor analyse the alpha or beta band. Of note is that many previous neuroimaging studies used very low numbers of participants, with the resulting APOE-ε4 carrier groups being typically below N = 20, with some notable exceptions in elderly participants (*Babiloni et al., 2006*; *de Waal et al., 2013*; *Sheline et al., 2010*). Our more highly-powered MEG study in young participants therefore provides valuable, reliable, and unique insights into the early effects of APOE-ε4 on oscillatory brain function.

Functional MRI (fMRI) resting-state studies typically show that the blood-oxygen-level-dependent (BOLD) response within the default-mode network (DMN) is reduced in healthy ageing (*Damoiseaux et al., 2008*). This reduction is accelerated in AD (*Buckner et al., 2008*) and is also modified in elderly APOE-ε4 carriers compared to non-carriers (*Sheline et al., 2010*). Filippini et al. (*Filippini et al., 2009a*) previously investigated DMN BOLD activity in a small sample of young APOE-ε4 carriers versus non-carriers (N = 18 each) using an independent component analysis (ICA) technique. They found increased co-activation of areas within the DMN for ε4 carriers, which were focused on hippocampal and frontal, but posterior regions were not significantly different. Furthermore, they also found an increase in the sensorimotor network. Despite the different spatial focus of effects, these fMRI findings are similar to ours in that they reveal increased brain connectivity in young APOE-ε4 carriers. In a more recent resting state-fMRI study of older, but healthy, APOE-ε4 carriers it was found that, as well as replicating Fillipini et al.'s finding of increased hippocampal DMN connectivity, significant increases in network activity were also found for APOE-ε4 carriers in posterior regions of the DMN, including the precuneus, and lateral parietal regions as well as frontal cortex (*Westlye et al., 2011*). Importantly, these frontal and posterior-lateral regions were only significantly increased in the right hemisphere (their *Figure 1D* and *Table 3*), showing good concordance with our findings of oscillatory hyperconnectivity in similar right-hemisphere regions. Finally, we note that some recent task-based fMRI studies have also shown APOE-ε4 carrier hyperactivity in posterior parts of the DMN (*Hodgetts et al., 2019*; *Shine et al., 2015*). Interestingly, the authors of these studies speculate that hyperactivity/hyperconnectivity of the DMN in these young APOE-ε4 carriers may reflect reduced network efficiency/flexibility.

Note that in our cohort, and other previous studies, there is little or no evidence of cortical volume changes, including hypothesised hippocampal volume decreases (*Filippini et al., 2009a*; *Heise et al., 2011*; *O'Dwyer et al., 2012*; *Wishart et al., 2006*), suggesting that changes in both haemodynamic and oscillatory brain function can be detected before changes in brain structure in the pathway of increased risk for AD (*Figure 5*, lower panel, (*Hampel et al., 2011*)).

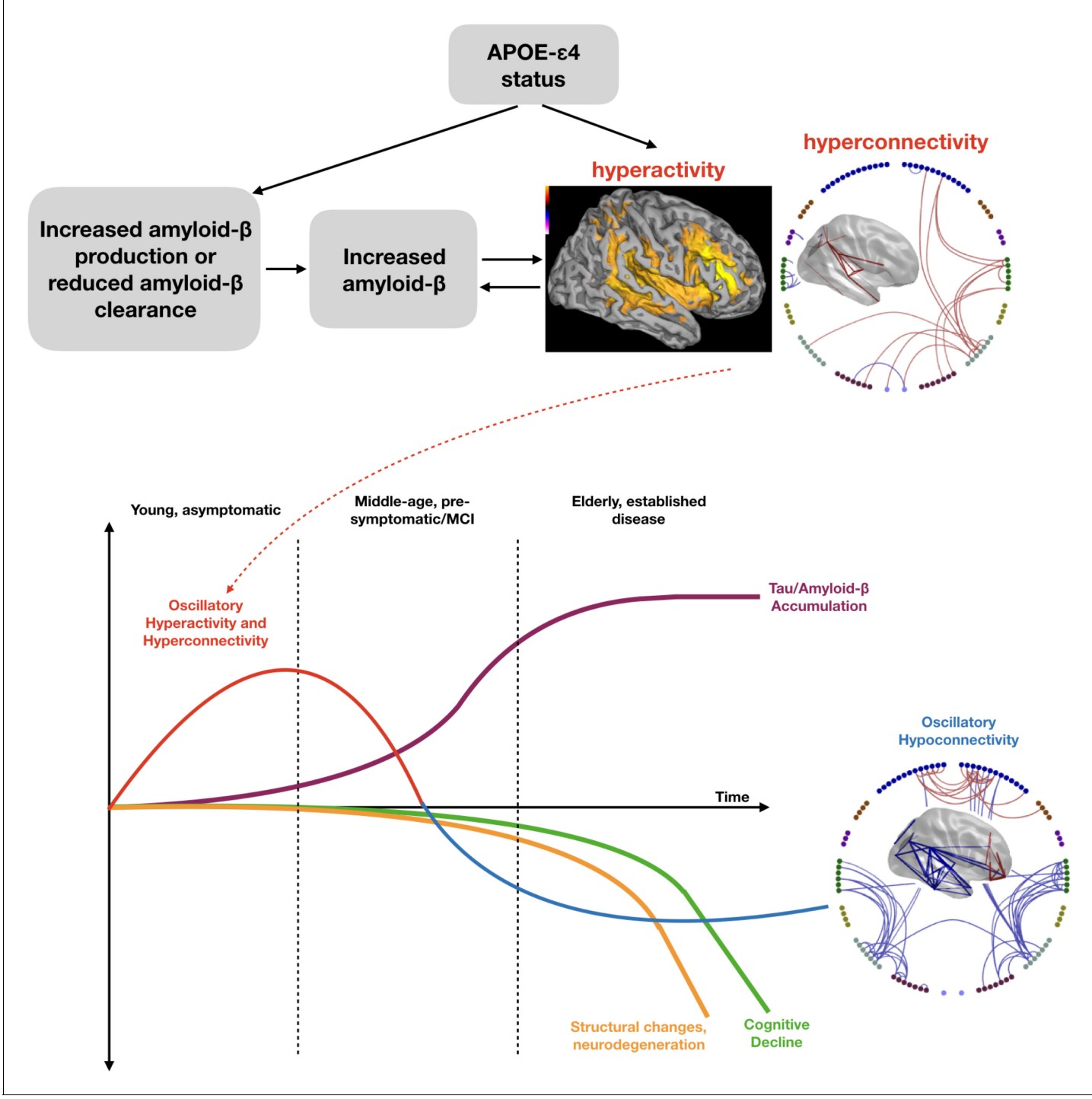

**Figure 5.** Schematic representation of a speculated mechanism (top tow) and order of events (bottom row) in brain structure and function when a healthy young individual develops Alzheimer's disease. Our results are consistent with theories that APOE-ε4 status may lead to early neuronal hyperactivity/hyperconnectivity, either directly via modification of the excitation/inhibition balance (*Nuriel et al., 2017*) or linked with amyloid deposition (*Stargardt et al., 2015*). Most biomarkers of disease show a progressive incline or decline, whereas our results suggest that functional connectivity shows a profile of an early increase, before structural, cognitive and neurobiological markers are evident. Eventually a more profound connectivity decrease is seen after a clinical diagnosis of Alzheimer's disease has been established. Inspired by *Hampel et al. (2011)*, *Figure 4*.
DOI: https://doi.org/10.7554/eLife.36011.009

The results presented here are, of course, only correlational in nature but we can speculate, in the presence of prior in-vitro and in-vivo animal studies, what underlying neurophysiological changes underpin our hyperactivity and hyperconnectivity findings. It is known that increased amyloid-β is somehow related to increased neural activity, although the sequence of cause and effect is not known (*Stargardt et al., 2015*). For example, it was shown in a rodent disease model that during Braak pre-symptomatic stages 0, I and II, amyloid-β starts to accumulate within neurons, whilst at the same time synaptic activity rises within the neurons (i.e. hyperactivity). As the disease progresses to Braak stage VI, and amyloid-β becomes significant enough to cause plaques, synaptic activity then falls (i.e. hypoactivity). Interestingly, it appeared that the rise in amyloid-β appeared to be linked to a reduction of activity in the Insulin-Degrading Enzyme (IDE).

However, what it is not clear is how *APOE-ε4* carrier status might lead to an increased risk of the hyperactivity <>amyloid-β relationship described above in young or even middle-aged people. Experimental studies have demonstrated that carrying the *APOE-ε4* allele leads to a wide range of effects in the brain, including impaired myelination, mitochondrial dysfunction, reduced cholinergic function and vascular pathology (*Reinvang et al., 2013*). Healthy elderly *APOE-ε4* carriers also show increased uptake of amyloid-binding PET ligands (PiB) (*Morris et al., 2010*; *Reiman et al., 2009*).

Already in middle age, post-mortem studies of *APOE-ε4* carriers have shown an increase in senile amyloid-β plaques (*Kok et al., 2009*), with some evidence of this even at ages as young as 30. It is therefore reasonable for us to speculate that in our young cohort, although amyloid-β plaques may not have yet developed, amyloid-β may be accumulating and, as in the rodent model, be associated with neural hyperactivity (*Figure 5*, upper panel). The mechanism for how *APOE-ε4* carrier status leads to increased amyloid-β deposition is complex and may involve multiple mechanisms, including disruptions of amyloid-β clearance systems dependent on receptor-mediated, cerebrovascular or proteolytic degradation (*Kanekiyo et al., 2014*; *Zhao et al., 2018*). For example, one study has shown that *APOE-ε4* leads to reduced levels of insulin-degrading enzyme (IDE), one of the most important enzymes for amyloid-β clearance (*Du et al., 2009*) and, as described above, is reduced during pre-symptomatic stages in rodent disease models which also show synaptic hyperactivity (*Stargardt et al., 2015*).

It has also been shown in a recent study, however, that neural hyperactivity can also occur in *APOE-ε4* mouse models in the absence of overt amyloid or tau pathology (*Nuriel et al., 2017*) via a mechanism in which *APOE-ε4* leads to an overall decrease in inhibition caused by a reduced responsiveness of excitatory neurons to GABAergic input. The authors therefore suggest that it is this *APOE-ε4* related shift in the excitation/inhibition balance that leads to hyperactivity that later results in increased amyloid deposition, and hence increased risk of AD.

It is important to note, though, that increased amyloid-β is not sufficient to cause AD (*Herrup, 2015*; *Sheline et al., 2010*) and the presence of *APOE-ε4* does not guarantee progression to AD. It remains to be investigated what role other important factors for AD play in this cascade of events, such as the tau protein (*Ossenkoppele et al., 2016*). Interestingly, PET studies of AD have also demonstrated AD-related hypometabolism in bilateral parieto-temporal regions that are very similar to the regions we report here (*Kato et al., 2016*).

Our findings therefore support theories based on rodent models (*Stargardt et al., 2015*), human computational models (*de Haan et al., 2012b*) and MCI data (*Maestú et al., 2015*) that propose that hyperactivity, ultimately related to hyperconnectivity, may be at the heart of brain dysfunction that could eventually lead to AD, and provides compelling evidence that this may already happen in early adulthood. Areas predisposed to hyperconnectivity are thought to be prone to yield increased accumulation of amyloid-β (*Myers et al., 2014*), supported by longitudinal PET and fMRI studies which found that hippocampal hyperactivation in elderly individuals leads to widespread increased deposition of amyloid-β as well as hippocampal signal loss and memory decline, although these studies found no effect of *APOE-ε4* (*O'Brien et al., 2010*) or there was insufficient power to test this (*Leal et al., 2017*). The present study suggests that the parietal cortex is an important common early focus for hyperconnectivity, possibly as part of the posterior DMN. Importantly, this hyperconnectivity is present during the task-free resting-state, both avoiding and highlighting the potential confound of baseline-level differences in task-based group studies.

One important methodological note, for future work, is that our exploratory analysis, in which we combined the connectivity maps from different frequency bands using a vector sum approach, revealed both more robust connectivity patterns across participants, with increased number of valid

edge connections, and increased sensitivity to both *APOE-ε4* carrier effects and Alzheimer disease-related changes in connectivity. This simple combination of effects may therefore be worth adding to future studies in order to, potentially, provide increased sensitivity to disease-related modulations.

In addition to classical statistics we used machine-learning classification approaches to investigate how much discriminative information is present within these static oscillatory connectivity matrices. Here, the support-vector-machine (SVM) was trained used a pooled set of features taken from all valid connectivity edges from all frequency bands and the combined map. The SVM will, as part of its decision boundary optimisation, choose the best weighting of all these features. When tested using leave-one-out cross validation, the classifier was able to decode *APOE-ε4* carriers from controls and AD patients from elderly controls, with significantly better than chance performance. In our small AD cohort, prediction performance appeared close to 70%, however, given the low number of participants in this study, this needs to be evaluated using a larger independent sample. Although our results here must be considered preliminary, they are encouraging – particularly as the AD patients in our study were relatively early in their diagnosis.

In the current study, we employed MEG to show patterns of oscillatory hyperactivity and hyper-connectivity in neural activity in young-adult *APOE-ε4* carriers, identifying the right parietal cortex as an important centre in these hyperconnectivity patterns. One particularly attractive feature of MEG is that it provides non-invasive high-temporal and spatial-resolution measures of neurophysiology that allow for comparison with cellular hyperactivity data in a translational context (*Busche et al., 2008*), providing potential biomarkers for new therapeutic and preventative studies in the pathological processes leading to AD. Finally, an important question is why and at what stage hyperconnectivity 'collapses' to convert to the profound disconnection and symptomology of AD, and what further factors determine whether ε4 carriers indeed go on to develop AD. Future studies focusing on the often-neglected middle-age group, between the very young and the elderly, as well as longitudinal studies will be invaluable to unravel the time course of these events.

## Limitations of the study

There are several methodological limitations of this study that need to be acknowledged. Firstly, all of the analyses we present here are associations or correlations than can only imply a link between APOE-ε4 status, hyperconnectivity and future risk of developing Alzheimer's disease – no causal relationships can be proved in this type of study. Similarly, the partial spatial overlap between the hyperconnectivity we observe in young APOE-ε4 carriers, compared to their control group, and hypoconnectivity shown in elderly AD patients, compared to their control group, is interesting but again does not demonstrate a causal link or a direct trajectory. Longitudinal studies that use our methods in individuals as they age, perhaps in older people with and without Mild Cognitive Impairment (MCI), to see if conversion to AD can be predicted would be an exciting extension of our work.

Although our young APOE-ε4 sample size is larger (51 versus 108) than much of the previous work in this field, it is still relatively small and our findings need replicating in an independent cohort. Such cohorts are starting to become available as online resources (e.g. CamCAN, http://www.cam-can.org/index.php?content=dataset) and it would be relatively straightforward to replicate our specific analysis pipeline, including our edge-thresholding procedure, to confirm the findings we report here. Second, as this is an exploratory study, we did not perform an exhaustive test of all connectivity metrics we could have employed, such as phase-based measures and dynamic connectivity analyses. We took the simplest possible measure – static oscillatory amplitude-amplitude envelope correlation as this has previously demonstrated the best within-participant reproducibility (*Colclough et al., 2016*) but this does not guarantee that such robust and repeatable measures are the most sensitive to cohort differences. It is entirely possible that other measures, particularly time-resolved dynamic connectivity analyses (*Vidaurre et al., 2016*), could be more sensitive and informative than the static amplitude-amplitude measure we used here.

Similarly, we took a conscious decision to only analyse connections that were assessed to be the strongest and thus, the most repeatable, across participants in our cohorts. In a way, this is a form of quality-control that rejects features (here connectivity edges) that are of low SNR and hence are poorly estimated in individuals and we note that such a procedure is common in the application of graph-theory methods for the analysis of network properties. It is entirely possible that we have

chosen a procedure, and in particular a threshold (mean rank of 80%), that is sub-optimal for revealing cohort differences. This is particularly relevant for MEG in which SNR is region-dependent and so we may have biased our analysis here to be more sensitive to superficial cortical regions (such as the parietal lobes) whilst suppressing an analysis of poor SNR regions, such as deeper regions in the brain. Future work could tune this parameter (as well as varying other parameters in our analysis pipeline) to try and maximise detection and discriminability of cohort effects across the brain. However, it is worth noting that MEG source-localisation algorithms, including the beamformer we use here, have non-uniform spatial resolution that means that regions with poor SNR, including deeper regions, are poorly resolved and hence activity is spread over wider regions. By restricting our statistical analyses to regions with higher SNR, we are protecting ourselves, somewhat, from localisation errors that could result from this spatial spreading.

In common with previous studies, we used a procedure in which, after trial-rejection and beamformer source localisation, trials were concatenated to form a continuous virtual-sensor timeseries for subsequent analysis. One potential problem with this is that it introduces discontinuities in the timeseries that could lead to sub-optimal estimations of connectivity. The precise impact of this is difficult to predict as it depends on the exact temporal distribution of rejected trials. However, it is likely to have the biggest impact on the very lowest frequencies, such as Delta, which only has relatively few oscillatory cycles in each trial. In our experiment there was no significant difference in the number of rejected trials between the groups (e.g. *APOE-ε*4 carriers and their controls) but this concatenation procedure may have reduced our sensitivity to effects in the Delta and Theta ranges, which showed relatively few robust connections compared to Alpha and Beta.

We chose to use a common anatomical parcellation, the AAL (*Tzourio-Mazoyer et al., 2002*) atlas, which ourselves and other groups (*Hillebrand et al., 2016*; *Hunt et al., 2016*; *Koelewijn et al., 2017*; *Pang et al., 2016*; *Routley et al., 2017*) have used in multiple MEG connectivity analysis papers, including our recent paper on oscillatory dysconnectivity in AD (*Koelewijn et al., 2017*). We therefore chose, a-priori, a commonly used atlas for our spatial reduction strategy. However, this atlas has 90 regions of largely varying size and shape and has been defined without regard to the sensitivity profile of our MEG system to different regions of the brain. We are therefore summarising activity, using a single virtual sensor, in what are large extended regions of widely varying SNR. This will bias our results to detecting connectivity in certain regions and may well be sub-optimal for detecting interactions between regions in which a finer, or coarser, parcellation may well have been beneficial. The principled way forward here might be to derive parcellations that use information regarding the MEG lead-field and the resulting resolution kernels in source-space (*Farahibozorg et al., 2018*).

A similar criticism might also be levelled at our source-leakage correction procedure in which we have used multivariate orthogonalisation (*Colclough et al., 2015*) simultaneously across all 90 timecourses to minimise all cross parcel correlations in the raw timeseries. This procedure may well be most successful in conditioning, and hence detecting, high-SNR signals which will tend to be more superficial signals with higher oscillatory power, such as the lateral parietal regions we describe in this paper. This differential sensitivity, which stems fundamentally from the relative lack of depth-sensitivity in MEG recordings but may be worsened by these analysis procedure choices, may well explain why our findings do not fully match those found in previous fMRI studies (*Filippini et al., 2009a*; *Westlye et al., 2011*), at least for deep and medial regions.

## Materials and methods

In this paper we present the results of analysing two MEG resting-state experiments: 1) A study of young healthy participants who have been genotyped for *APOE-ε*4 and 2) A study of Alzheimer's disease (AD) patients compared to a matched cohort of elderly controls.

### Participants in experiment 1: *young APOE-ε4 cohort*

The *APOE* analysis was performed on resting-state MEG data acquired at CUBRIC as part of the '100 Brains' and 'UK MEG Partnership' projects, which together yielded a total of 183 usable datasets. Inclusion criteria were aged 18–65, completed or be undertaking a degree, normal or corrected-to-normal vision, and no history of neurological or neuropsychiatric disorder. Of the 183 participants, 51 participants were identified as heterozygote or homozygote *APOE-ε*4 carriers, and

108 as not carrying an ε4 allele (demographic details in *Table 1*). Genotyping was unsuccessful in the remaining 24 participants, who were excluded from analysis. All participants gave informed consent according to the Declaration of Helsinki and all procedures were approved by the Ethics Committee of the School of Psychology, Cardiff University (EC.12.01.10.3071).

### *APOE* genotyping in Experiment 1

Genomic DNA was obtained from saliva using Oragene OG-500 saliva kits (DNA Genotek, Inc., Ontario, Canada). Genotyping was performed using custom Illumina HumanCoreExome-24 Bead-Chip genotyping arrays, which contain approximately 500,000 common genetic variants (Illumina, Inc., San Diego, CA). Quality control and imputation were implemented in plink 1.9 (*Chang et al., 2015*). Individuals were excluded for any of the following reasons: 1) ambiguous sex (where samples with undetermined X chromosome heterozygosity were excluded); 2) cryptic relatedness up to third-degree relatives as ascertained using identity by descent; 3) genotyping completeness less than 98%; 4) non-European ethnicity admixture which was determined via population stratification, where samples that clustered outside the CEU HapMap3 population using principal component analysis were excluded; and 5) outliers from an autosomal heterozygosity filter. Single nucleotide polymorphisms (SNPs) were excluded where the minor allele frequency was less than 1%, if the call rate was less than 98%, or if the $\chi 2$ test for Hardy-Weinberg equilibrium had a p value less than 1e-6. A total of 233,054 genotyped SNPs remained after quality control. Autosomal chromosomes were imputed using the reference panel HRCv1.1 (hrc.r1.1.2016) using a mixed population panel (*McCarthy et al., 2016*). Phasing was completed using Eagle v2.3 (*Loh et al., 2016*) and imputation was performed using Mimimac3 (*Das et al., 2016*). Imputed data was converted to best-guess genotypes (*Chang et al., 2015*) with multi-allelic sites removed. SNP filters for HWE (1e-6) and minor allele frequency (1%) were re-applied. A total of 7,545,595 SNPs were successfully imputed. Variants rs7412 and rs429358 were used to estimate APOE genotype for all 183 individuals (*Corneveaux et al., 2010*).

### *Experiment 1 young APOE-ε4 cohort: data acquisition and analysis*

#### Data acquisition

Five-minute whole-head MEG recordings were acquired at a 1200 Hz sample rate using a 275-channel CTF radial gradiometer system. An additional 29 reference channels were recorded for noise cancellation purposes and the primary sensors were analysed as synthetic third-order gradiometers (*Vrba and Robinson, 2001*). Subjects were seated upright in the magnetically shielded room with their head supported with a chin rest to minimise movement. Participants were asked to rest and fixate their eyes on a central red fixation point, presented on either a CRT monitor or LCD projector. Horizontal and vertical electro-oculograms (EOG) were recorded to monitor eye blinks and eye movements.

Participants also underwent a magnetic resonance imaging (MRI) session in which a T1-weighted 1 mm anatomical scan was acquired, using an inversion recovery spoiled gradient echo acquisition (3T, General Electric).

Participants further performed the MATRICS Consensus Cognitive Battery (MCCB) (*Kern et al., 2008*; *Nuechterlein et al., 2008*). Group performance was compared using two-sample t-tests, excluding missing or incomplete data (leaving per task N = 47–48 for ε4 carriers and N = 98–100 for non-carriers, see *Table 1*).

#### MEG data analysis

MEG generates multi-dimensional data, which can be analysed in a large variety of ways. We sought to conduct an analysis investigating only consistent functional connectivity across the brain, while reducing noise. We focused on amplitude-amplitude connectivity of beamformer-derived oscillatory source signals, one of the most robust and repeatable types of MEG connectivity measures (*Colclough et al., 2016*), across six frequency bands and 90 AAL atlas brain areas. The analysis pipeline also allowed a voxelwise assessment of resting-state activity within each frequency band and is displayed in *Figure 6*.

All datasets were down-sampled to 600 Hz, and filtered with a 1 Hz high-pass and a 150 Hz low-pass filter. The datasets were then segmented into 2 s epochs. The data were visually inspected and

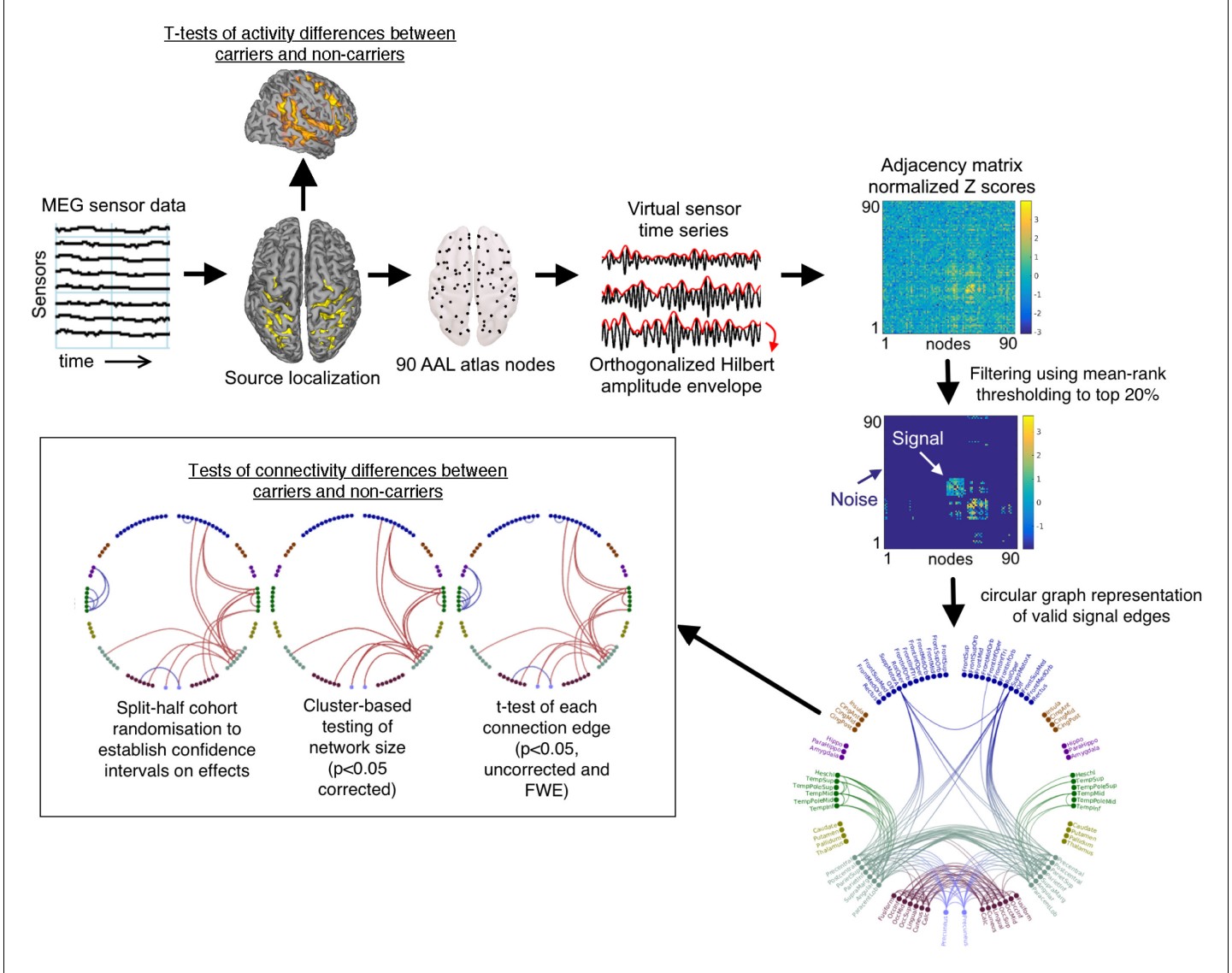

**Figure 6.** Schematic overview of the analysis pipeline. Note that in the labelled circular plot (bottom right) the nodes are in approximate anatomical locations as if the brain was viewed from above (left is left/right is right) and are coloured according to their originating lobes (Blue: frontal, Gold: Insula/Anterior-Posterior cingulate, Purple: Medial temporal, Green: Temporal: Teal: Parietal and Sensorimotor, Maroon: occipital, Lilac: Precuneus. When statistical effects are plotted, red represents cohort increases and blue cohort decreases.

DOI: https://doi.org/10.7554/eLife.36011.010

epochs with major artefacts such as head movements, large muscle contractions, or ocular artefacts were excluded from subsequent analysis.

To achieve MRI/MEG co-registration, fiduciary markers were placed at fixed distances from three anatomical landmarks identifiable in the subject's anatomical MR scan, and their locations were manually marked in the MR image. Head localization was performed at the start and end of each recording.

The MEG sensor data were source-localised using FieldTrip (RRID:SCR_004849) version 20161011 (*Oostenveld et al., 2011*), with an LCMV beamformer on a 6 mm grid, using a single-shell forward model (*Nolte, 2003*), where the covariance matrix was constructed per each of the following frequency bands: 1–4, 3–8, 8–13, 13–30, 40–60, and 60–140 Hz. For each band, the beamformer weights were normalized using a vector norm (*Hillebrand et al., 2012*), data were normalized to the MNI template, and reduced to 90 nodes following the Automatic Anatomical Labelling (AAL) atlas

(*Tzourio-Mazoyer et al., 2002*). The selection of one node per AAL region was achieved by performing a frequency analysis on all virtual channels within the AAL region, and selecting the virtual channel with the greatest temporal standard-deviation within the region. Epochs were concatenated to generate a continuous virtual-sensor timecourse for each voxel and then band-passed into the relevant frequency-band of interest (Delta: 1–4 Hz, Theta: 3–8 Hz, Alpha: 8–13 Hz, Beta-13–30 Hz, Low Gamma: 40–60 Hz and High-Gamma: 60–140 Hz).

The resulting 90-node time series were orthogonalized to avoid spurious correlations using a multivariate regression approach known as symmetric orthogonalization (*Colclough et al., 2015*). The data then underwent a Hilbert transform to obtain the oscillatory amplitude envelope, and were subsequently despiked to remove artefactual temporal transients using a median filter, downsampled to 1 Hz and trimmed to avoid edge effects (removing the first 2 and last three samples). Amplitude correlations were then calculated by correlating the 90 downsampled Hilbert envelopes to each other, and converted to variance-normalized Z-scores by applying a Fisher transform. In addition to analysing effects within each frequency band separately, we also synthesised a combined measure of connectivity by calculating the vector-sum of all connectivity matrices, using the following formula for each element (i,j) in the square connectivity matrix:

$$\text{Combined}_{ij} = \text{sqrt}(\text{Delta}_{ij}{}^2 + \text{Theta}_{ij}{}^2 + \text{Alpha}_{ij}{}^2 + \text{Beta}_{ij}{}^2 + \text{LowGamma}_{ij}{}^2 + \text{HighGamma}_{ij}{}^2)$$

## Assessment of activity differences

In order to assess possible cohort differences in oscillatory activity, rather than connectivity, we also calculated a measure of temporal variance at each beamformer-reconstructed voxel in the brain, for each separate frequency band, via an assessment of the amplitude envelopes. Due to large variations in signal sensitivity throughout the brain, beamformer reconstructions of virtual-sensors result in signals in which the mean and temporal standard-deviation of the amplitude envelopes are highly correlated and show large variations across the source space. For that reason, we choose to use a normalized measure, the coefficient of variation, which is the simply the standard-deviation of the amplitude envelope divided by the mean of the amplitude envelope. Using this measure, voxels which show a high fractional temporal variability, compared to the mean, will have a high value, whilst those voxels that are of more stable oscillatory power will have a low value. We note that the measure is similar, by analogy, to common measures employed in BOLD fMRI, such as percentage-change from baseline. The statistical significance of group-effects was assessed using univariate voxelwise unpaired t-tests of this activity measure and then correcting for multiple comparisons using randomisation testing using FSL *randomise* (*Nichols and Holmes, 2003*; *Singh et al., 2003*; *Winkler et al., 2014*). The latter procedure was performed using threshold-free cluster enhancement (TFCE) and 5000 iterations, without variance smoothing. This procedure tests positive and negative cohort contrasts separately so a null hypothesis corrected p-value of 0.025 was chosen, so as to implement a corrected two-tailed test at p<0.05. Separately for each frequency band, thresholded t-statistics are displayed on a template brain surface mesh, extracted using *FreeSurfer*.

## Statistical analysis of group differences in connectivity

First, each participant's connectivity matrix was z-scored to have zero mean and unit variance across all connections, in order to correct for possible global differences, due to variability in data quality, either across participants or cohorts (*Siems et al., 2016*). All subsequent analyses are therefore assessments, at each connection edge, of the relative strength of connectivity, compared to the mean, in each participant. This procedure has the advantage of ensuring that any systematic differences, either at the participant or cohort-level, are controlled for.

To avoid analysing noise, we then independently assessed which functional connections were consistently present across all 159 datasets, for each frequency band and in the Combined maps. This was done by calculating the rank of every connection in each participant's connectivity map and then averaging this map rank across all participants. The resulting average rank-map was thresholded at 80%, such that we only consider as 'valid' these connectivity edges that are consistently in the top 20%. Importantly, to ensure that large cohort differences are not missed, we performed this assessment of valid connections separately for each cohort, and labelled connections as valid if they passed our rank-mask threshold in either cohort. The remaining connections were discarded.

We statistically tested the difference between groups using three different statistical approaches. In the first, we used unpaired t-tests of the corrected Z-scores, at each of the 'valid' connectivity edges, and looked for significant edges at both p<0.05 (uncorrected) and at p<0.05 (corrected using a 5000-permutation test with omnibus thresholding).

Secondly, we tested for significant differences in connected network cluster size using methods similar to those described in the NBS toolbox (*Zalesky et al., 2010*). Here, connection differences are first thresholded at p<0.05 (uncorrected) before Matlab's (RRID:SCR_001622) *graph* function is used to convert this difference map to a binarized graph. Matlab's *conncomp* function is then used to find the largest connected cluster within this graph. The corrected significance of this largest cluster is assessed using randomisation testing with 5000 iterations and thresholded at p<0.05 (corrected). This network-based statistic has previously been shown to be more sensitive to cohort effects than the simple mass-univariate testing we use in our first approach (*Zalesky et al., 2010*). As previously described, we test positive and negative cluster sizes separately.

Thirdly, we tested the likely generalisation of our findings by assessing the confidence interval on cohort-related increases and decreases in connectivity. This was done by randomly sampling half the *APOE-ε4* carriers (i.e. N = 26) and half the non- *APOE-ε4* carriers (i.e. N = 54) and for each of the valid edges in the map we tabulate whether the mean edge connectivity is increased or decreased in the *APOE-ε4* group. This sampling strategy is repeated 5000 times and those edges that do not have a consistent sign – either increases or decreases – on more than 95% of samples are discarded. The end result is a set of connections that are robustly increased, or decreased, in the *APOE-ε4* carriers compared to non-carriers, irrespective of the specific individuals entered into the test.

This is an exploratory study that used previously collected data to search for differences related to *APOE-ε4* carrier status. To our knowledge there has not been another published MEG study in young volunteers but we can estimate statistical power based on other studies. Power was estimated using the *pwr* function in R (*Champely et al., 2018*). The area under the curve (AUC) for *APOE-ε4* status, calculated in a recent Alzheimer's disease case-control study (*Escott-Price et al., 2015*) was transformed to Cohen's *d* (*Ruscio, 2008*), yielding a moderate effect size (*d* = 0.653). For this effect size, for our sample of 51 carriers and 108 non-carriers, we have approximately 87% power to detect an effect of APOE-ε4 status on our measures.

All network plots and brain connectivity diagrams were generated using custom in-house scripts in Matlab (*Shaw, 2019*).

## Supervised Machine-Learning analysis

An alternative way of investigating whether the static oscillatory connectivity matrices contain information that distinguishes cases, compared to controls, is to attempt to train a classifier that can distinguish between the two groups. Here we used a support vector machine (SVM) (*Boser et al., 1992*) analysis in which we took all valid edges, pooled across all estimated maps for each person (Delta, Theta, Alpha, Beta, Low Gamma, High Gamma, Combined) and used these as features for the SVM. Note that, in order to prevent biased performance, features were selected by taking the top 20% of edges, estimated over the whole cohort so that features were not selected separately from each group. In machine-learning it is important to use cross-validation to prevent over-fitting (as we have more features than participants). Ideally this would be done by training on one dataset and testing on another, but here we only have one dataset. We therefore used a Leave-One-Out (LOO) sampling approach in which we randomly select one case (i.e. an *APOE-ε4* carrier) and one control (i.e. a non-carrier) as our test sample and hence train the classifier using all other participants. We then test how well this classifier performs on the two test participants we held back. This procedure is repeated 5000 times to build up an estimate of how well the classifier can successfully categorise each participant. We then assess the accuracy of the classifier by pooling these out-of-training estimates to obtain mean measures of Sensitivity (i.e. the true-positive rate), Specificity (true-negative rate) and the area under the receiver operating characteristic curve (AUC of the ROC). For this binary classification the chance value for all of three measures is 0.5. One important note is that, for the *APOE-ε4* experiment, we have a significantly unbalanced classification problem in that there are twice as many non-carriers as carriers. This is important because the classifier could achieve a performance of 66% by simply labelling all participants as non-carriers. We guard against this by taking a conservative approach of balancing the classification, at each iteration, by only using a random sample of 50 non-carriers and 50-carriers to train the classifier and, in addition, reporting the three

evaluation metrics we describe above. Finally, the performance of all classification analyses is ultimately assessed using the AUC. To assess the statistical significance compared to chance, we use random permutation (50,000 iterations) of the true labels to build an empirically estimated null distribution of the AUC and then tested the achieved AUC against this.

## Cortical volume analysis

To investigate whether brain structure was affected in *APOE*-ε4 carriers, we performed cortical segmentation using FreeSurfer v6.0 (RRID:SCR_001847, surfer.nmr.mgh.harvard.edu). We calculated hippocampal volumes per hemisphere and total intracranial volume (ICV). Relative hippocampal volumes were calculated by dividing hippocampal volumes by ICV. Following quality control, performed using a protocol from ENIGMA (http://enigma.ini.usc.edu/) (*Stein et al., 2012*), volumes were excluded for 1 ε4 carrier and two non-carriers. Using the same data and analysis procedure, cortical thickness, volume and area were also assessed within each of the subdivisions of the Desikan-Killiany atlas (*Desikan et al., 2006*) included within FreeSurfer.

## Alzheimer's disease and elderly control cohort: MEG acquisition and analysis

Recruitment, acquisition and pre-processing of the AD patients (N = 14, 5F/9M, mean age 77.7 years, Mini-Mental State Examination (MMSE) score 19–27, all fulfilled criteria for probable AD [*McKhann et al., 1984*]) and age-matched controls (N = 11, 7F/4M, mean age 74.4 years, MMSE score 28–30) has been described in detail previously (*Koelewijn et al., 2017*). In summary though, for inclusion as probable AD, criteria were that dementia had been established by clinical and cognitive assessment (ACE-R), that cognitive impairments were progressive and included episodic memory deficit and deficit in at least one other area of cognition, and there was no evidence of other factors that could explain the dementia syndrome such as cerebrovascular disease identified from a previous CT scan. The study was approved by the South-East Wales Research Ethics Committee (10/WSE04/35), and all participants provided informed consent. The procedure was identical to the *APOE* analysis, except that participants were asked to rest while opening and closing their eyes following an auditory verbal cue (24 × 15 s alternations in total). Each dataset was split into one eyes-open and one eyes-closed dataset, following which the statistical analysis of AD patients, compared to the matched elderly control sample, proceeded in exactly the same way as that described above for the *APOE*-ε4 study in young controls. Here, we report comparison results only for the eyes-open resting data. This was done in order to compare the pattern of effects revealed by comparing young *APOE*-ε4 carriers to non-carriers with those obtained by comparing AD participants to an age-matched sample of elderly controls in the same resting-state paradigm. Note that we have previously reported results from this AD-control comparison for eyes-open resting state using a different analysis pipeline (*Koelewijn et al., 2017*) but, in that dataset, 14/16 AD patients and 11/21 controls viewed a visual grating instead of a simple blank screen. In this analysis, we decided to only include those participants who saw a visual grating during the eyes-open period, in order to make sure that patient-related effects were not due to presence, or not, of the visual grating as this was not matched across the cohorts. This means we have a reduced subset of those analysed previously that is N = 14 AD patients and N = 11 elderly controls who all performed an identical version of the resting-state paradigm.

## Acknowledgements

The authors would like to thank Mark Woolrich and Giles Colclough for providing the 'ROInets' package code for orthogonalisation and variance-normalization of the connectivity Z-scores, and Alexander Shaw for help with atemplate visualisations. We would also like to thank Suresh Muthukumaraswamy for initial versions of the AAL template code.

# Additional information

## Funding

| Funder | Grant reference number | Author |
|---|---|---|
| BRACE | | Loes Koelewijn<br>Aline Bompas<br>Andrea Tales<br>Antony Bayer<br>Krish Singh |
| Medical Research Council | MR/K004360/1 | Thomas M Lancaster<br>David Linden<br>Krish Singh |
| Medical Research Council | MR/K501086/1 | Bethany C Routley |
| Medical Research Council | MR/K005464/1 | Bethany C Routley<br>Lorenzo Magazzini<br>Krish Singh |
| Wellcome | WT105613/Z/14/Z | Katherine E Tansey |

The funders had no role in study design, data collection and interpretation, or the decision to submit the work for publication.

## Author contributions

Loes Koelewijn, Formal analysis, Validation, Visualization, Methodology, Writing—original draft, Writing—review and editing; Thomas M Lancaster, Data curation, Formal analysis, Validation, Investigation, Writing—review and editing; David Linden, Conceptualization, Resources, Data curation, Supervision, Funding acquisition, Project administration, Writing—review and editing; Diana C Dima, Methodology; Bethany C Routley, Lorenzo Magazzini, Kali Barawi, Lisa Brindley, Aline Bompas, Data curation, Investigation, Project administration; Rachael Adams, Investigation, Project administration; Katherine E Tansey, Resources, Methodology; Andrea Tales, Antony Bayer, Resources; Krish Singh, Conceptualization, Resources, Software, Supervision, Funding acquisition, Validation, Visualization, Methodology, Project administration, Writing—review and editing

## Author ORCIDs

Loes Koelewijn [iD] http://orcid.org/0000-0002-7890-171X
Diana C Dima [iD] http://orcid.org/0000-0002-9612-5574
Lorenzo Magazzini [iD] https://orcid.org/0000-0002-8934-8374
Katherine E Tansey [iD] http://orcid.org/0000-0002-9663-3376
Krish Singh [iD] https://orcid.org/0000-0002-3094-2475

## Ethics

Human subjects: All participants gave informed consent according to the Declaration of Helsinki and all procedures were approved by the Ethics Committee of the School of Psychology, Cardiff University (EC.12.01.10.3071) and the South Wales NHS Research Ethics Committee (10/WSE04/35).

## Decision letter and Author response

Decision letter https://doi.org/10.7554/eLife.36011.015
Author response https://doi.org/10.7554/eLife.36011.016

# Additional files

## Supplementary files

• Transparent reporting form
DOI: https://doi.org/10.7554/eLife.36011.011

## Data availability

Both the necessary derived data and custom scripts used to create Figures 1, 2, 3 and 4 are freely available as an Open-Science Framework project (http://dx.doi.org/10.17605/OSF.IO/E4CJX). Due to the sensitive data of the genetic risk factors investigated here, and the need to prevent both self-identification and third-party identification of people in this project, both the raw data and pseudo-anonymised intermediate analysis results cannot be provided. Controlled access to the pseudo-ano-nymised intermediate analysis results might be possible via a research collaboration after signing of suitable data-transfer agreements and approval by both Cardiff University School of Psychology's Ethics Committee and the governing body for the cohort, i.e. the National Centre for Mental Health (NCMH). Further enquiries should be directed to Professor Krish D Singh at CUBRIC, School of Psy-chology, Maindy Road, Cardiff University, Cardiff CF24 4HQ or email: singhkd@cardiff.ac.uk.

The following dataset was generated:

| Author(s) | Year | Dataset title | Dataset URL | Database and Identifier |
|-----------|------|---------------|-------------|--------------------------|
| Singh KD | 2019 | Derived data/measures and Matlab code used to create the figures | http://dx.doi.org/10.17605/OSF.IO/E4CJX | Open Science Framework, 10.17605/OSF.IO/E4CJX |

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
