## [Decision Letter]

Thank you for submitting your article "Disrupted parietal oscillatory connectivity links young *APOE-*ɛ4 carriers to Alzheimer's disease" for consideration by *eLife*. Your article has been reviewed by David Van Essen as the Senior Editor, a Reviewing Editor, and three reviewers. The following individuals involved in review of your submission have agreed to reveal their identity: Hanna Renvall (Reviewer #1); Allison Nugent (Reviewer #2); Annika Hulten (Reviewer #3).

The reviewers have discussed the reviews with one another, and the Reviewing Editor has drafted this decision to help you prepare a revised submission.

Summary:

The study by Koelewijn et al., involves a data set of 183 subjects in an attempt to link APOE-ɛ4 carriers to preclinical changes neural activity, in relation to their increased genetic risk of developing AD. The authors investigated resting-state oscillatory network connectivity across different frequency bands. Key findings: APOE-ɛ4 carries have increased alpha- and beta-band connectivity in bilateral occipital areas and parietal cortices, as compared to non-carriers. The connectivity changes were partially overlapping with decreased connectivity between right hemisphere parietal and temporal areas in a separate population of AD patient. Based on these findings the authors suggest that AD can be characterized by pre-onset hyperconnectivity that, once the disease manifests, turns into hypoconnectivity particularly in the parietal cortex.

In general, the referees were positive and found the study exciting. However, in particular in regard to methodology there were several concerns that require clarifications and justification. Also, it was a general feeling that the Discussion section requires improvement: in its current form the paper lacks deeper discussion and theoretical background needed to interpret the present findings with respect to the majority of the existing literature in AD, as well as on the possible clinical value of the results.

One referee asks for further analysis (see point 14): In the description of the AD subjects, the manuscript states that there was an age-matched cohort. However, it appears they were not compared to the AD subjects. It is proposed to use these as a comparison group rather than the young E4 carriers. The authors are encouraged to follow this advice or provide a strong rationale for why this was not done.

Essential revisions:

Methodology

1) It is unclear why the authors decided to use only eyes open condition here (cf. their earlier study on AD patients with both eyes open and closed data), and not e.g. the difference between eyes closed (with prominent alpha activity) and eyes open conditions that might be a more reliable functional measure? Furthermore, it appears that the differences between AD patients and controls were even more prominent in the eyes closed condition? Please motivate.

2) The section on the Gaussian Mixture Model (GMM) is inadequately described. The authors state that a GMM was applied to all 8100 connection to separate signal from noise. Within each frequency band, however, aren't there only 4005 unique connections, since correlations cannot discern directionality? How many subpopulations were included in the mixture model? Is there any published validation that shows that this method effectively separates true correlations from noise? Was this done for each group separately?

3) Is the discarding of connections +/- two scaled median SD's after the connections were removed by the GMM method? So, each subject now has a different number of valid connections?

4. The authors state that the valid connection masks were added to form a single mask – was this a union or an intersection? It says that it included all valid connections present in either group, but does this include connections that were not valid in all subjects?

5) How were the randomization test (akin to a two-sample t-test) and the uncorrected two sample t-test different, other than that one is parametric and the other non-parametric? The authors seem to indicate that the interpretation is different. (i.e. the two-sample t-test can test whether differences were global across the brain).

6) Please clarify how the power was calculated, what parameters were used to connect the effect size for disease prediction accuracy (the AUC from Escott-Price) to the MEG measures?

7) It is stated that the GMM selected valid datasets as well as valid connections, although the GMM methods do not mention how datasets were defined as valid or invalid.

8) One referee mentions that a main concern is the choice of parcellation scheme used in the connectivity analysis. While it is fair to say that there exists no one correct atlas that always should be used, the AAL is definitely non-optimal for a MEG connectivity analysis as the parcels are of very different size and shape. By selecting only one node from each parcel, the nodes will be unevenly distributed across the cortex, which in itself is somewhat problematic. However, a bigger problem is that large elongate parcels (such as those in the temporal lobe) may contain two (or more) active areas but will be represented by only one node (capturing the strongest source). The worst-case scenario is if a strong active area is situated on the border between two or more large parcels. In this case the nodes from each of these parcels will reflect the same underlying neural activity, whereas a clear but somewhat smaller activation in the other end of one of the parcels would go undetected. Assuming that the multivariate leaking correction works as intended, two nodes capturing the same activity is less of a problem (though it may lead to spatially incorrect inferences) but ignoring an active area may alter the network dynamics radically. The most straight forward way around this is to select a parcellation scheme with optimally sized parcels (not too small, not too big) that are of roughly equal size. To my knowledge no existing atlas fulfills these criteria fully, which is probably reflected in the fact that most connectivity papers use modified versions of existing parcellations. If the number of parcels becomes too big, there is the option to select only those that cover the cortical regions associated with the DMN in the literature. Alternatively, you may wish to try a more sophisticated approach and apply a data driven way of finding all active regions based on some form of clustering algorithm or ICA, and the define parcels/nodes based on these. Please comment on the AAL approach in relation to these alternatives.

9) The use of a multivariate orthogonalization approach is an elegant way to address the problem of spurious (aka ghost) interactions. However, like all methods it has its limitations, and it would be good to acknowledge these in the Discussion section. For example, the SNR is not constant across all cortical regions in MEG and areas which will the source estimate specificity and by extension the rotation done in the correction. Therefore, connectivity between areas with high source power (like the occipital and parietal regions in the alpha band) may be more easily detected than those with low source power (like the frontal cortex). This may, in part be the reason for the different results between the previous fMRI study by Filippini et al.

Results/interpretation

10) The major finding suggest hyperactivity in the right parietal connectivity, but after the statistical analysis the result appears quite modest and very local. How can one be sure that it is not the result of signal leakage?

11) The location for the statistically significant hyperactivity is not anatomically the most evident one, taken that the most prominent structural changes appear in the hippocampus and more generally in the medial temporal lobes. The authors have analyzed the hippocampal volumes and total intracranial volumes in all subjects, but what about the parietal volumes? Is their e.g. any hemispheric difference in the parietal lobe volumes in the APOE-ɛ4 carriers that would correlate with the present finding?

12) In Table 2, the only overlapping, affected areas between non-symptomatic APOE-ɛ4 carriers and AD patients were the right parietal areas, whereas 6/10 nodes with greatest group difference in APOE-ɛ4 carriers and controls were not different between AD patients and age-matched controls. How do the authors explain this disappearing difference along with the progressing disease?

13) In Figure 2, from panels D and H, it appears that all connections were greater in the E4 carriers compared to non-carriers. In panel E, it is clear that most of the valid connections are above the x=y line, indicating that the mean Z is higher for the carriers than non-carriers. However, in panel A, it appears that most of the valid connections are below the x=y line, which would indicate that the mean Z is larger in the non-carriers. Please clarify. Also, in panes F and G, it appears that there are valid connections to frontal areas, however these are not apparent on the glass brains.

14) (also see general comment for this point). The description of participants is confusing, given that a re-analysis of data in AD was performed, but these subjects are not well described. Please clarify. There also seems to be a comparison between young E4 carriers and AD patients: I question the utility of this given that age is a confound. In the description of the AD subjects the manuscript states that there was an age-matched cohort, but I do not see where they were compared to AD subjects. This would be a much better comparison group than young E4 carriers.

15) Table 2 shows nodes most "affected" within each group separately – what is being compared? Eyes open vs. eyes closed? Based upon Figure 4, is it right that the comparisons are young APOE-E4 vs. AD, and AD vs. matched controls? Why are the results for E4 carriers vs. non-carriers in Figure 4 different from the results presented earlier?

Discussion

16) In its current form the paper lacks deeper discussion and theoretical background needed to interpret the present findings with respect to the majority of the existing literature in AD, as well as on the possible clinical value of the results.

17) The AD patients who are APOE-ɛ4 carriers are known to develop the clinical AD at an earlier stage than non-carriers, but in the end with very similar structural and clinical changes to the non-carriers. Why this happens, and what is the role of APOE-ɛ4 allele in the course of the disease remains unclear, but it has nevertheless been studied extensively. The manuscript should provide some light on what is known about the cellular-level functions of APOE-ɛ4, and especially on how these functions might be related to the present neuroimaging results. I do see that it is difficult to draw conclusions between cellular level findings and non-invasive neuroimaging results, but at least some hypothesis/discussion would be needed here.

18) Taken that the ultimate goal is to find individual measures on the connectivity changes in the prodromal patients, the decision to concentrate only on the connections that were consistently present in all datasets would benefit some comment/discussion on the selection's potential drawbacks.

19) PET often shows hypometabolic changes already in the stage of mild cognitive impairment in AD patients – is there literature on PET results in young healthy APOE-ɛ4 carriers? If such exists, it should be discussed with respect to the present results.

20) Taken that MEG may not be the optimal method to study deep brain structures or frontal cortex, some comment on the possible limitations of MEG in this respect would be good to add here.

21) When searching for clinical biomarkers, the consistency of the measures over multiple measurements should be demonstrated. Any such information available to be added here?

22) In discussing both their own findings and the previous literature, the authors should avoid inferring causation from correlation. This line of reasoning occurs both in the introduction and discussion. For example, in the Introduction the authors state that the previous findings "suggest that the presence of an APOE-ɛ4 allele alters oscillatory brain function", when in fact the studies that are referred to have only found an association. The fact that the presence APOE-ɛ4 does not always lead to AD (with accompanying changes in oscillatory brain activity) suggests that other genetic or environmental factors (such as e.g. cholesterol levels) are at play as well. Moreover, it is possible (if not probable) that the APOE-ɛ4 is part of a more multifaceted genetic basis of AD. Assigning the causation to one allele would in this case be misleading.

23) A referee was unable to follow the logic in the Discussion section where the authors compare the present findings to those reported by Filippini et al. The author claim that the previous fMRI study corroborates the present findings even though completely different areas are highlighted in each of the respective studies. I am also uncertain what the word "inconsistency" is referring to in the sentence "In contrast, evidence of the hippocampal volume decrease in young carriers is inconsistent […]". You referring to previous findings being inconsistent or that the present study adds to the inconsistency? I suspect that the authors mean that given the inconsistent findings of reduced hippocampal measures, hemodynamic and/or oscillatory network effect may be better markers of an increased AD risk. Still given that there are only 2 network studies even this seems a bit of an overstatement. In either case, please clarify this paragraph.

[Editors' note: further revisions were requested prior to acceptance, as described below.]

Thank you for resubmitting your work entitled "Oscillatory hyperactivity and hyperconnectivity in young *APOE*-ɛ4 carriers and hypoconnectivity in Alzheimer's disease" for further consideration at *eLife*. Your revised article has been favorably evaluated by Huda Zoghbi as the Senior Editor, a Reviewing Editor, and three reviewers.

The manuscript has been improved but there are some remaining issues that need to be addressed before acceptance, as outlined below:

The reviews below provide details about the limitations that need to be addressed. In summary, all reviewers agree that revision of the text (especially the claims) and figures is still needed. Please revise the text putting special emphasis on the limitation/claim parts and on the details of the methodology. While we agreed that no additional analyses need to be done, the epoching of the time series should be mentioned, perhaps in limitations.

I hope that the comments below will help you revised the manuscript.

*Reviewer #1:*

The authors have evidently gone through a big effort in re-analyzing the data and in replying to the reviewer comments on the earlier manuscript. I think that the machine learning approach gives a significant contribution to the analyses.

As to the present manuscript, I would still be careful in using words like "prevention" from which the present data is still quite far. Providing possible predictive information on AD in very early stage is already an important step.

In general, the language is not always best possible and could still be clarified and polished. Also, the amount of given details on the data analysis continues to vary, which would make it difficult to replicate part of the applied analyses: e.g. the analysis based on Graph Theory (subsection “Statistical Analysis of Group Differences in Connectivity”) mainly refer to existing toolboxes, while the SVM approach is rather extensively explained.

*Reviewer #2:*

The revised paper is significantly improved in terms of clarity of the methods and results. While many of my concerns have been addressed, the number of revisions performed by the authors have prompted a few additional comments:

I have a concern about the division of the time series into 2 second epochs, removal of segments with artifacts, followed by concatenation. While the number of retained epochs were consistent across groups, what about the number of discontinuities introduced into the time series? For example, removing two adjacent epochs would result in one discontinuity, while removing two separate epochs would result in two discontinuities. What is the impact of this on the connectivity? This may be particularly problematic for the lowest frequency band; there will only be 2-8 cycles of the 1-4 Hz band. I am not suggesting that the analyses be re-done, although some commentary on this point may be warranted.

Why was a visual grating used rather than a more "pure" resting state study?

In Figure 1 – I'm not sure how relevant row 2 is, given that I would expect many connections to survive a p<0.05 uncorrected threshold by chance alone. This is especially true since row 2 is quite similar to row 4. Also, in this figure, given the narrow width of the lines, I can't visually detect any differences in opacity. Perhaps line width would be a better indicator of connection strength?

In Figure 2 - I'm not entirely clear on the utility of the lower panel. Although there is some overlap with the areas showing higher gamma activity and areas showing greater connectivity, it is not entirely convincing that these two things are related in any way, given that more regions that don't overlap than regions that do.

Figure 3 – same comment as for Figure 1.

Figure 4 – it may be useful to show which specific connections do overlap, perhaps a conjunction of the graphs shown in the bottom panel of 4A. Is there a negative correlation between the strengths of the connections that are in this conjunction? How many edges actually overlap?

I am concerned that the results are somewhat overstated in the conclusions. There was at best partial overlap between the oscillatory power and connectivity results. Likewise, there was only partial overlap between the connections found in the young sample and the older sample.

Some comment on the regions showing the greatest abnormality may be helpful to put the results in context.

*Reviewer #3:*

This is a follow-up review on the study by Koelewijn et al. that looks at changes in the oscillatory connectivity in resting state MEG in APOE-ɛ4 carriers and matched controls. The authors have, in my opinion replied to the reviewers’ comments and suggestions in a satisfactory manner, and I am ready to suggest that the manuscript be accepted for publication in *eLife*.

Before signing off on this review, I would nonetheless like to express that I am somewhat disappointed that the authors, contrary to my suggestion, chose not to change the parcellation template used in the connectivity analysis. I am sympathetic to the authors' argument that the AAL is widely used, but this does mean that it's a good one. Most likely the reason, that is widely used is merely a consequence of the methodological dominance of fMRI where the template is much less problematic. As the authors themselves note in their Discussion, it remains a fact that the AAL is not well-suited for connectivity analysis of MEG data. Bad practices in science do not change unless someone leads the way. Hence, I encourage the authors to at least in their future studies become trail makers that help change suboptimal practices, and thus advancing also the validity of MEG.

---

## [Author Response]

Summary:The study by Koelewijn et al., involves a data set of 183 subjects in an attempt to link APOE-ɛ4 carriers to preclinical changes neural activity, in relation to their increased genetic risk of developing AD. The authors investigated resting-state oscillatory network connectivity across different frequency bands. Key findings: APOE-ɛ4 carries have increased α- and β-band connectivity in bilateral occipital areas and parietal cortices, as compared to non-carriers. The connectivity changes were partially overlapping with decreased connectivity between right hemisphere parietal and temporal areas in a separate population of AD patient. Based on these findings the authors suggest that AD can be characterized by pre-onset hyperconnectivity that, once the disease manifests, turns into hypoconnectivity particularly in the parietal cortex.In general, the referees were positive and found the study exciting. However, in particular in regard to methodology there were several concerns that require clarifications and justification. Also, it was a general feeling that the Discussion section requires improvement: in its current form the paper lacks deeper discussion and theoretical background needed to interpret the present findings with respect to the majority of the existing literature in AD, as well as on the possible clinical value of the results.One referee asks for further analysis (see point 14): In the description of the AD subjects, the manuscript states that there was an age-matched cohort. However, it appears they were not compared to the AD subjects. It is proposed to use these as a comparison group rather than the young E4 carriers. The authors are encouraged to follow this advice or provide a strong rationale for why this was not done.

There seems to have been a misunderstanding here, and we apologise for not explaining this properly. We did two experimental comparisons. (1) Young APOE-ɛ4 carriers were compared to Young non-carriers and (2) Elderly AD participants were compared to an age-matched control group. We do not compare the elderly group to the young group as this would be dominated by age-related changes. We do, however, directly compare the statistical effects calculated separately for the two experimental comparisons (see Figure 5).

Essential revisions:Methodology1) It is unclear why the authors decided to use only eyes open condition here (cf. their earlier study on AD patients with both eyes open and closed data), and not e.g. the difference between eyes closed (with prominent α activity) and eyes open conditions that might be a more reliable functional measure? Furthermore, it appears that the differences between AD patients and controls were even more prominent in the eyes closed condition? Please motivate.

The APOE study is an opportunistic one in which we have post-hoc analysed data from an internal database at Cardiff called “the 100 Brains study”. In this study, the resting-state recording was one of many recordings performed in the MEG, in addition to many further tests and neuroimaging modalities. It was a practical decision at the time of designing this study to only collect eyes-open data to reduce the total time required. As a result, we only have eyes-open data available for this analysis. Furthermore, we actually found a very similar pattern in eyes-open and closed resting-state recordings in our previous paper (Koelewijn et al., 2017), which has also been reported by others (Hillebrand et al., 2016). For this reason, we do not think that results would have been much different had we presented eyes-closed data, or a difference between eyes-open and eyes-closed. Unfortunately, we do not have the data to perform this analysis and as such we restrict our findings to eyes-open resting-state for both experiments. To avoid confusion, the eyes-closed analysis (which was only available for the AD versus elderly control experiment) has been removed from the manuscript.

2) The section on the Gaussian Mixture Model (GMM) is inadequately described. The authors state that a GMM was applied to all 8100 connection to separate signal from noise. Within each frequency band, however, aren't there only 4005 unique connections, since correlations cannot discern directionality? How many subpopulations were included in the mixture model? Is there any published validation that shows that this method effectively separates true correlations from noise? Was this done for each group separately?

We have decided not to use this relatively complex (and new) approach but instead use a simple rank-thresholding procedure. Namely, to take those connections that have a Mean Rank across the cohort of >=80% i.e. the strongest 20% connections. For Alpha and Beta this identified an extremely similar pattern of connections as the GMM approach, but also now reveals high-rank correlations in Delta, Theta and High Gamma.

3) Is the discarding of connections +/- two scaled median SD's after the connections were removed by the GMM method? So, each subject now has a different number of valid connections?

Actually, we were completely removing participants from the analysis, if they had too many or two few valid connections. As we describe above, we have now removed this procedure.

4. The authors state that the valid connection masks were added to form a single mask – was this a union or an intersection? It says that it included all valid connections present in either group, but does this include connections that were not valid in all subjects?

It was a union i.e. a logical ‘OR’ approach. In this, connections identified as ‘valid’ in either group were included to avoid missing out on potential group differences. Otherwise, if a connection was very greatly increased/decreased in one cohort compared to another, the connection would potentially be discarded, and a large group difference missed. Note that connections are accepted/discarded at the group level by assessing their average mean rank across the cohort – not at the single-subject level.

5) How were the randomization test (akin to a two-sample t-test) and the uncorrected two sample t-test different, other than that one is parametric and the other non-parametric? The authors seem to indicate that the interpretation is different. (i.e. the two-sample t-test can test whether differences were global across the brain).

The uncorrected two-sample t test, we believe, is a useful first look at the distribution of connection strength differences across the brain. The randomisation test, in which we use omnibus testing to derive the null distribution for the largest difference in the brain, estimates whether each of those connections would survive a multiple comparison correction i.e. it is simply used to assess corrected p-values. We use this method instead of Bonferroni correction across all 4010 unique non-diagonal correlation pairs because it is less conservative in the presence of known non-independence of the measures see: (Nichols and Holmes, 2003) (Singh, Barnes and Hillebrand, 2003)

6) Please clarify how the power was calculated, what parameters were used to connect the effect size for disease prediction accuracy (the AUC from Escott-Price) to the MEG measures?

This was not used in the MEG measures, only to assess the potential statistical power in the genetics study.

7) It is stated that the GMM selected valid datasets as well as valid connections, although the GMM methods do not mention how datasets were defined as valid or invalid.

Please see our response to Point 3, above.

8) One referee mentions that a main concern is the choice of parcellation scheme used in the connectivity analysis. While it is fair to say that there exists no one correct atlas that always should be used, the AAL is definitely non-optimal for a MEG connectivity analysis as the parcels are of very different size and shape. By selecting only one node from each parcel, the nodes will be unevenly distributed across the cortex, which in itself is somewhat problematic. However, a bigger problem is that large elongate parcels (such as those in the temporal lobe) may contain two (or more) active areas but will be represented by only one node (capturing the strongest source). The worst-case scenario is if a strong active area is situated on the border between two or more large parcels. In this case the nodes from each of these parcels will reflect the same underlying neural activity, whereas a clear but somewhat smaller activation in the other end of one of the parcels would go undetected. Assuming that the multivariate leaking correction works as intended, two nodes capturing the same activity is less of a problem (though it may lead to spatially incorrect inferences) but ignoring an active area may alter the network dynamics radically. The most straight forward way around this is to select a parcellation scheme with optimally sized parcels (not too small, not too big) that are of roughly equal size. To my knowledge no existing atlas fulfills these criteria fully, which is probably reflected in the fact that most connectivity papers use modified versions of existing parcellations. If the number of parcels becomes too big, there is the option to select only those that cover the cortical regions associated with the DMN in the literature. Alternatively, you may wish to try a more sophisticated approach and apply a data driven way of finding all active regions based on some form of clustering algorithm or ICA, and the define parcels/nodes based on these. Please comment on the AAL approach in relation to these alternatives.

This is a very good set of points and we thank the reviewer for these. The AAL is an increasingly used, almost default, atlas in this area of research but, of course, we have to acknowledge these potential problems. With that in mind, we have added a new subsection “Limitations of the study”. The reviewer will see we have added a section on this specific issue to acknowledge it and to point to a potential way forward.

9) The use of a multivariate orthogonalization approach is an elegant way to address the problem of spurious (aka ghost) interactions. However, like all methods it has its limitations, and it would be good to acknowledge these in the discussion. For example, the SNR is not constant across all cortical regions in MEG and areas which will the source estimate specificity and by extension the rotation done in the correction. Therefore, connectivity between areas with high source power (like the occipital and parietal regions in the α band) may be more easily detected than those with low source power (like the frontal cortex). This may, in part be the reason for the different results between the previous fMRI study by Filippini et al.

Again, this is a very important point and so we have added another paragraph describing this, to the end of the subsection “Limitations of the study”.

Results/interpretation10) The major finding suggest hyperactivity in the right parietal connectivity, but after the statistical analysis the result appears quite modest and very local. How can one be sure that it is not the result of signal leakage?

In the prior draft of the paper, the reviewer is quite right that after statistical correction at individual edges only one edge remained in the Beta band. In fact, when we revised our analysis procedure (see start of this document) this connection dropped below our significance criteria, whereas a single connection in the theta band now becomes significant. Put simply, with the effect sizes we have, and a conservative correction for multiple comparisons, we do not seem to see extensive connectivity differences at the level of individual edges. However, we have augmented our reporting by also assessing connected cluster size and now report this and we also use cohort-resampling to see how reproducible effects are across sub-samples of our participants. These do seem to suggest robust patterns of statistical effects – just not at the individual edge level.

11) The location for the statistically significant hyperactivity is not anatomically the most evident one, taken that the most prominent structural changes appear in the hippocampus and more generally in the medial temporal lobes. The authors have analyzed the hippocampal volumes and total intracranial volumes in all subjects, but what about the parietal volumes? Is their e.g. any hemispheric difference in the parietal lobe volumes in the APOE-ɛ4 carriers that would correlate with the present finding?

This is an excellent suggestion and we have now added an analysis of regional thickness, volume and area to the manuscript, using the standard Desikan-Killiany gyral atlas within Freesurfer. No significant differences in any parameter were found in any of the regions, after correction for multiple region comparisons.

12) In Table 2, the only overlapping, affected areas between non-symptomatic APOE-ɛ4 carriers and AD patients were the right parietal areas, whereas 6/10 nodes with greatest group difference in APOE-ɛ4 carriers and controls were not different between AD patients and age-matched controls. How do the authors explain this disappearing difference along with the progressing disease?

In the revised manuscript we make it clear that some of the effects are overlapping, but there is not a clear one to one correspondence. We have decided to illustrate this visually with multiple visualisations in the new Figure 5 and have removed the table describing individual connections. We believe that the scatter plot shown in new Figure 5 shows that there is a general inverse relationship between connectivity differences observed in Young APOE-ɛ4 carriers (compared to young controls) and AD patients (compared to their age-matched cohort). The plots of the networks themselves, though reveal that the effects are spatially overlapping but not perfectly registered. This is now explained in the text, figures and this paragraph:

“Our results therefore suggest that the right parietal cortex is a particularly important area in the risk for, and development of, AD. Although our study can only show an implied association between early hyperconnectivity and later hypoconnectivity, we could speculate that an initial hyperconnectivity/hyperactivity centred around (right) parietal cortex has a cascade of effects that may lead to eventual disconnection in AD. Although initially a right-dominant effect, this may spread over time to more widespread effects across the cortex and hemispheres. A larger study, using the methods we present here, over a wider age-range, may shed further light on this.”

13) In Figure 2, from panes D and H, it appears that all connections were greater in the E4 carriers compared to non-carriers. In panel E, it is clear that most of the valid connections are above the x=y line, indicating that the mean Z is higher for the carriers than non-carriers. However, in panel A, it appears that most of the valid connections are below the x=y line, which would indicate that the mean Z is larger in the non-carriers. Please clarify. Also, in panes F and G, it appears that there are valid connections to frontal areas, however these are not apparent on the glass brains.

Apologies – the reviewer was correct. We made an error in the cohort resampling procedure so that connections surviving our confidence-interval criteria were always shown in red – even if they were robust decreases. For the Alpha band, the posterior connections should have been blue, reflecting decreases in the APOE-ɛ4 group. This has been fixed in the new analyses.

Also note that the two frontal connections displayed in panels F and G were the right 7rolandic operculum and supplementary motor area. These two areas are positioned quite posterior in the frontal cortex, and therefore appear close together to the various parieto-temporal areas displayed in the glass brain.

14) (also see general comment for this point). The description of participants is confusing, given that a re-analysis of data in AD was performed, but these subjects are not well described. Please clarify. There also seems to be a comparison between young E4 carriers and AD patients: I question the utility of this given that age is a confound. In the description of the AD subjects the manuscript states that there was an age-matched cohort, but I do not see where they were compared to AD subjects. This would be a much better comparison group than young E4 carriers.

Sorry again for the misunderstanding – Please see our opening comments, but in short, we never directly compared young APOE-ɛ4 carriers to AD patients. Statistical comparisons were always to an age-matched control group in both cases. What we do perform (See new Figure 5) is a comparison of both the spatial distribution and magnitude of (APOE-ɛ4-young controls) and (AD-elderly control) effects.

15) Table 2 shows nodes most "affected" within each group separately – what is being compared? Eyes open vs. eyes closed? Based upon Figure 4, is it right that the comparisons are young APOE-E4 vs. AD, and AD vs. matched controls?

Please see our previous comments and we have tried hard in the redrafted manuscript to make this clearer.

Why are the results for E4 carriers vs. non-carriers in Figure 4 different from the results presented earlier?

In the redrafted manuscript it should be clearer that these are now the same.

Discussion16) In its current form the paper lacks deeper discussion and theoretical background needed to interpret the present findings with respect to the majority of the existing literature in AD, as well as on the possible clinical value of the results.This is a very good suggestion and we have now extended the Introduction, Discussion section (and redrafted Figure 6) to try and address this.17) The AD patients who are APOE-ɛ4 carriers are known to develop the clinical AD at an earlier stage than non-carriers, but in the end with very similar structural and clinical changes to the non-carriers. Why this happens, and what is the role of APOE-ɛ4 allele in the course of the disease remains unclear, but it has nevertheless been studied extensively. The manuscript should provide some light on what is known about the cellular-level functions of APOE-ɛ4, and especially on how these functions might be related to the present neuroimaging results. I do see that it is difficult to draw conclusions between cellular level findings and non-invasive neuroimaging results, but at least some hypothesis/discussion would be needed here.

Please see the new extended Introduction and the Discussion section. As the reviewer probably can understand this is a very complex area, so we have tried to synthesise major points and point readers to major review articles.

18) Taken that the ultimate goal is to find individual measures on the connectivity changes in the prodromal patients, the decision to concentrate only on the connections that were consistently present in all datasets would benefit some comment/discussion on the selection's potential drawbacks.

Please see the new subsection “Limitations of the study” in which we describe the rationale for this decision and why it may lead to a relative insensitivity to some cohort effects. In general, as this is an exploratory study describing a new finding, it is probably best to be on the conservative side.

19) PET often shows hypometabolic changes already in the stage of mild cognitive impairment in AD patients – is there literature on PET results in young healthy APOE-ɛ4 carriers? If such exists, it should be discussed with respect to the present results.

As far as we have been able to find, no such PET studies have been performed in young healthy APOE-ɛ4 carriers.

20) Taken that MEG may not be the optimal method to study deep brain structures or frontal cortex, some comment on the possible limitations of MEG in this respect would be good to add here.

Please see the new subsection “Limitations of the study” in which we make this clear.

21) When searching for clinical biomarkers, the consistency of the measures over multiple measurements should be demonstrated. Any such information available to be added here?

We are only aware of one published paper showing the repeatability of our connectivity measures (Colclough et al., 2016) and this is cited now in the Materials and methods section:

“We focused on amplitude-amplitude connectivity of beamformer-derived oscillatory source signals, one of the most robust and repeatable types of MEG connectivity measures (Colclough et al., 2016)”.

And we also discuss/cite this in the new subsection “Limitations of the study” – It is our prime motivation for only using amplitude-amplitude connectivity measures.

22) In discussing both their own findings and the previous literature, the authors should avoid inferring causation from correlation. This line of reasoning occurs both in the introduction and discussion. For example, in the Introduction the authors state that the previous findings "suggest that the presence of an APOE-ɛ4 allele alters oscillatory brain function", when in fact the studies that are referred to have only found an association. The fact that the presence APOE-ɛ4 does not always lead to AD (with accompanying changes in oscillatory brain activity) suggests that other genetic or environmental factors (such as e.g. cholesterol levels) are at play as well. Moreover, it is possible (if not probable) that the APOE-ɛ4 is part of a more multifaceted genetic basis of AD. Assigning the causation to one allele would in this case be misleading.

Yes – this is, again, a very good point. We have been through the manuscript and toned down our implication of a direct link, including the sentence mentioned, which now reads:

“These findings suggest that the presence of an APOE-ɛ4 allele is associated with differences in oscillatory brain function at a stage preceding AD symptomology, but it is not clear how early in the life span this is established”.

We have also added a section to the subsection “Limitations of the study” making it explicit that this type of study only reveals associations, not causal relationships.

23) A referee was unable to follow the logic in the Discussion section where the authors compare the present findings to those reported by Filippini et al. The author claim that the previous fMRI study corroborates the present findings even though completely different areas are highlighted in each of the respective studies. I am also uncertain what the word "inconsistency" is referring to in the sentence "In contrast, evidence of the hippocampal volume decrease in young carriers is inconsistent […]". You referring to previous findings being inconsistent or that the present study adds to the inconsistency? I suspect that the authors mean that given the inconsistent findings of reduced hippocampal measures, hemodynamic and/or oscillatory network effect may be better markers of an increased AD risk. Still given that there are only 2 network studies even this seems a bit of an overstatement. In either case, please clarify this paragraph.

We thank the reviewer for pointing this out. We have redrafted this entire section (Discussion section), but also added a further study from Westlye et al., 2011. Although in an older group than ours (and Fillipini’s) Westlye also found similar increased activation in the hippocampal parts of the DMN for E4 carriers, but importantly for us, also found increased DMN synchronisation in the posterior and medial parts of the DMN. Interestingly for the lateral cortical areas, including parietal, they also found an exclusively right hemisphere increase in DMN synchronisation for APOE-4 carriers, matching quite nicely with our findings.

[Editors' note: further revisions were requested prior to acceptance, as described below.]

The reviews below provide details about the limitations that need to be addressed. In summary, all reviewers agree that revision of the text (especially the claims) and figures is still needed. Please revise the text putting special emphasis on the limitation/claim parts and on the details of the methodology. While we agreed that no additional analyses need to be done, the epoching of the time series should be mentioned, perhaps in limitations.I hope that the comments below will help you revised the manuscript.

We have carefully reviewed the text and softened some of the statements. A paragraph detailing the potential problem with epoch concatenation has been added to the Limitations section of the Discussion. We have also added sub-figures to Figure 4 and a new Table 2 showing a conjunction analysis (requested by reviewer 2).

There are two other changes that we have made:

1) As discussed below in response to reviewer 1, we reviewed our cross-validation strategy for the SVM classification and realised that it needed amendment. Put simply, we were choosing the strongest 20% of edges as classification features for the SVM separately for the two groups being tested and then pooling all these edges for classification. We assumed that this was un-biased as we were not choosing features that maximised discrimination but, of course, in a real-world situation, this could not be done. Furthermore, to our surprise, using further randomisation testing confirmed that this did result in >50% performance for completely randomly-labelled data in many cases. So, we changed the procedure to select the top 20% of features using the whole cohort. This resolved the bias, but results in reduced performance, albeit still significantly greater than chance. We have therefore changed the results in Table 3 (previously Table 2) to reflect and moderated our discussion points regarding this analysis.

2) Unfortunately, when double-checking our results and packaging them up for open release (see below), we realised that the code used to generate Figure 3 (the AD cohort results) and the right column of Figure 4a had the wrong threshold set. When we re-ran the corrected script, the differences were subtle and for the presented data are hard to spot (it mostly is apparent in the count of significant edges shown on each plot i.e. in the last revision, the bottom right circle plot has 19/56 edges whereas now it has 23/56 edges. Note that no other figure, or analysis is affected by this error – it was purely a display problem. However, in addition, we now find 11 edges in the Low Gamma range, so have added the relevant column to Figure 3. The derived results and Matlab scripts needed to generate Figure 1, the lower part of Figure 2, Figure 3 and Figure 4 are now publicly released on an Open Science Framework project (https://osf.io/e4cjx/ or DOI: 10.17605/OSF.IO/E4CJX) so readers can independently verify our plots, or indeed change the statistical procedures or display parameters to further investigate the results. Please see our modified data availability statement in the manuscript as well.

Reviewer #1:The authors have evidently gone through a big effort in re-analyzing the data and in replying to the reviewer comments on the earlier manuscript. I think that the machine learning approach gives a significant contribution to the analyses.

Many thanks for this. Unfortunately, as can be seen in Table 3, we have re-evaluated our cross-validation procedure (now described in more detail in our Materials and methods section) and believe our original performance measures were inflated. The results are still significantly above chance but are not as high as previously reported. With that in mind we have altered the text describing the ML results to make it much clearer that these are encouraging results but moderate in effect.

As to the present manuscript, I would still be careful in using words like "prevention" from which the present data is still quite far. Providing possible predictive information on AD in very early stage is already an important step.In general, the language is not always best possible and could still be clarified and polished.

We’ve deleted the final part of the sentence in the Introduction that mentions prevention strategies.

Also, the amount of given details on the data analysis continues to vary, which would make it difficult to replicate part of the applied analyses: e.g. the analysis based on Graph Theory (subsection “Statistical Analysis of Group Differences in Connectivity”) mainly refer to existing toolboxes, while the SVM approach is rather extensively explained.

Actually, we don’t use an existing toolbox here but a very simple procedure in Matlab. We’ve deleted the reference to “Graph-Theory” (as we think this is confusing) and made it clear how the largest connected cluster size is estimated in subsection “Statistical Analysis of Group Differences in Connectivity”. This would allow the reader to replicate this procedure quite easily, including the subsequent statistical testing. We have left in the reference to the NBS toolbox as the Authors were the first to propose this method and also demonstrated its increased sensitivity.

Reviewer #2:The revised paper is significantly improved in terms of clarity of the methods and results. While many of my concerns have been addressed, the number of revisions performed by the authors have prompted a few additional comments:I have a concern about the division of the time series into 2 second epochs, removal of segments with artifacts, followed by concatenation. While the number of retained epochs were consistent across groups, what about the number of discontinuities introduced into the time series? For example, removing two adjacent epochs would result in one discontinuity, while removing two separate epochs would result in two discontinuities. What is the impact of this on the connectivity? This may be particularly problematic for the lowest frequency band; there will only be 2-8 cycles of the 1-4 Hz band. I am not suggesting that the analyses be re-done, although some commentary on this point may be warranted.

Thank you. This is a very import comment and entirely justified in requiring a comment. We have added a section on this to the subsection “Limitations of the study”.

Why was a visual grating used rather than a more "pure" resting state study?

Unfortunately, this was simply the design of the original study. As we mention in the paper, we are re-analysing the data from our first paper (Koelewijn et al.).

In Figure 1 (Figure 3 – same comment as for Figure 1), I'm not sure how relevant row 2 is, given that I would expect many connections to survive a p<0.05 uncorrected threshold by chance alone. This is especially true since row 2 is quite similar to row 4.

Row 2 and Row 4 are expressing different statistical analyses. Row 2 is a test for any effect in the difference of means, between cohorts, using a t-test, whereas Row 4 shows the consistency of the sign of the effect at each connection using randomised split-half testing of the cohort. The reviewer is right that for a large sample of normally distributed values these two things should converge, so it is actually quite encouraging that the maps are so similar – but that is not necessarily guaranteed, so we would prefer to show both results. Finally note that for the smaller cohort in the AD experiment there is more divergence between Row 2 and Row 4.

Also, in this figure, given the narrow width of the lines, I can't visually detect any differences in opacity. Perhaps line width would be a better indicator of connection strength?

This is potentially a good idea, but when we tried this the figure become rather confusing as the thick lines prevented viewing of other connections, particularly in the small versions of these circle plots. It is also worth noting that because many of these maps are thresholded, the variability of the effect magnitude is relatively small and hence the opacity does not vary much. For example, we only show the confidence interval values on Row 4 from 0.95 to 1.0.

In Figure 2 – I'm not entirely clear on the utility of the lower panel. Although there is some overlap with the areas showing higher γ activity and areas showing greater connectivity, it is not entirely convincing that these two things are related in any way, given that more regions that don't overlap than regions that do.

We actually quite like the lower panel as it allows the reader to easily identify (i) Which AAL nodes show hyperconnectivity in E4 carriers, (ii) Which AAL nodes show hypoconnectivity in E4 carriers, (iii) Which nodes show hyperactivity in E4 carriers. This is very difficult to do from the other Figures. The reviewer is absolutely right though, that the overlap is partial. However, note that those nodes in the right hemisphere that show hyperconnectivity also show hyperactivity (but not vice-versa). We have therefore moderated the claims in subsection “*APOE*-ɛ4 carrier versus non-carrier group differences in oscillatory activity Q” i.e.:

“In summary, gamma hyperactivities were found in mostly right-lateralised regions, some of which also appear to show hyperconnectivity in the lower frequency ranges (principally beta and alpha). This overlap is described graphically in the lower part of Figure 2.”

Figure 4 – it may be useful to show which specific connections do overlap, perhaps a conjunction of the graphs shown in the bottom panel of 4A. Is there a negative correlation between the strengths of the connections that are in this conjunction? How many edges actually overlap?

We have now added this plot to the bottom of 4A as requested, added an unthresholded conjunction plot to 4C and created a Table describing the effect sizes for these conjunction edges in a new Table 2. We’ve also added more details in the Figure caption.

I am concerned that the results are somewhat overstated in the conclusions. There was at best partial overlap between the oscillatory power and connectivity results. Likewise, there was only partial overlap between the connections found in the young sample and the older sample.

This is a good point. We have now changed the statements that discuss overlap to make it clear that the overlap is only partial between power and connectivity. Similarly, for the connectivity patterns in the two experiments. We have now added a table that shows that overlap in connections and the overlap in nodes showing connectivity differences.

Some comment on the regions showing the greatest abnormality may be helpful to put the results in context.

These are described in the new Table 2.

Reviewer #3:This is a follow-up review on the study by Koelewijn et al. that looks at changes in the oscillatory connectivity in resting state MEG in APOE-ɛ4 carriers and matched controls. The authors have, in my opinion replied to the reviewers comments and suggestions in a satisfactory manner, and I am ready to suggest that the manuscript be accepted for publication in eLife. However, there are a few minor language issues that they should fix (see below), to make the text more readable and suitable for a high impact publication.Before signing off on this review, I would nonetheless like to express that I am somewhat disappointed that the authors, contrary to my suggestion, chose not to change the parcellation template used in the connectivity analysis. I am sympathetic to the authors' argument that the AAL is widely used, but this does mean that it's a good one. Most likely the reason, that is widely used is merely a consequence of the methodological dominance of fMRI where the template is much less problematic. As the authors themselves note in their Discussion, it remains a fact that the AAL is not well-suited for connectivity analysis of MEG data. Bad practices in science do not change unless someone leads the way. Hence, I encourage the authors to at least in their future studies become trail makers that help change suboptimal practises, and thus advancing also the validity of MEG.

Many thanks for this, accepted and taken on board.